# Prioritizing non-coding regions based on human genomic constraint and sequence context with deep learning

Dimitrios Vitsios[1✉], Ryan S. Dhindsa[1], Lawrence Middleton[1], Ayal B. Gussow[2] & Slavé Petrovski[1✉]

Elucidating functionality in non-coding regions is a key challenge in human genomics. It has been shown that intolerance to variation of coding and proximal non-coding sequence is a strong predictor of human disease relevance. Here, we integrate intolerance to variation, functional genomic annotations and primary genomic sequence to build JARVIS: a comprehensive deep learning model to prioritize non-coding regions, outperforming other human lineage-specific scores. Despite being agnostic to evolutionary conservation, JARVIS performs comparably or outperforms conservation-based scores in classifying pathogenic single-nucleotide and structural variants. In constructing JARVIS, we introduce the genome-wide residual variation intolerance score (gwRVIS), applying a sliding-window approach to whole genome sequencing data from 62,784 individuals. gwRVIS distinguishes Mendelian disease genes from more tolerant CCDS regions and highlights ultra-conserved non-coding elements as the most intolerant regions in the human genome. Both JARVIS and gwRVIS capture previously inaccessible human-lineage constraint information and will enhance our understanding of the non-coding genome.

[1] Centre for Genomics Research, Discovery Sciences, BioPharmaceuticals R&D, AstraZeneca, Cambridge, UK. [2] National Center for Biotechnology Information, National Library of Medicine, Bethesda, MD, USA. ✉email: dimitrios.vitsios@astrazeneca.com; slav.petrovski@astrazeneca.com

The growing collection of human whole-genome sequencing data has allowed researchers to identify stretches of the genome that are preferentially intolerant to genetic variation. The resulting statistics equip geneticists to identify parts of our genome with the greatest potential to cause disease when mutated. For the protein-coding component of the human genome, we now have multiple metrics that capture disease potential at the level of the gene[1,2] and regions within a gene[3–5] with high confidence. These scores have transformed our ability to identify disease-causing mutations in the exome[1,6]. However, the majority of human genetic variation resides in non protein-coding regions of the genome[7,8], and our ability to interpret variants has been limited because the functional importance of these regions is largely unknown. Improved understanding of the relationship between noncoding variation and clinical disease can therefore provide a more comprehensive understanding of disease biology and reveal opportunities for the development of novel therapeutic targets.

Early studies have attempted to introduce methods that assess intolerance to a mutation in the noncoding genome to improve our understanding of variation in these regions. While these metrics have shown promise[6,9,10], their resolution has been limited due to small sample sizes of whole-genome sequencing (WGS) reference cohorts and may be biased due to strong dependence on evolutionary conservation in addition to human constraint in constructing noncoding intolerance scores. However, regulatory elements have high evolutionary turnover[11,12], which can obfuscate the use of conservation to interpret variation for many regions in the noncoding genome. The increasing sizes of WGS reference cohorts now offer an opportunity to assess intraspecies human variation at an even higher resolution.

Here, we introduce intolerance metrics that examine regions of the noncoding genome that may be purged of extensive variation due to purifying selection within the human lineage, adopting a larger reference set and a machine learning-based approach. We have previously introduced ncRVIS[6], which quantifies constraint in proximal non-coding regions such as promoters and untranslated regions. We first expand this method to the entire genome using a sliding window approach (with single-nucleotide resolution) to create a genome-wide residual variation intolerance score (gwRVIS). We then integrate genome-wide intolerance with information on primary genomic sequence and additional functional genomic annotation to build a comprehensive pathogenicity prediction framework for noncoding variants in the human genome. We intentionally do not employ any conservation information for the construction of our scores. This allows us to pinpoint regions that are more likely to be human-specific in terms of their functional relevance and provide a complementary human-lineage score to the many established phylogenetic conservation-based scores. Our metrics aim to facilitate prioritizing regions among the noncoding human genome which when mutated may be more likely to correlate with a clinically relevant effect.

## Results

We first sought to construct a score that captures the genome-wide intolerance to variation profile. We applied a tiled gwRVIS to WGS data from 62,784 individuals available in the TOPMed dataset[13] (Freeze5 release). The original RVIS score[1] quantifies intolerance by regressing the number of observed common functional (missense and protein-truncating) variants on the total number of observed variants in a gene. The resulting regression line predicts the expected number of common functional variants, and the deviation of each gene from this expectation (more or less variation than expected) is calculated as the residual divided by an estimate of its standard deviation (studentized residual).

In the genome-wide approach, we no longer have pre-defined genomic units (like genes in RVIS) or functional annotations for variants. Thus, we scan the entire genome with a sliding-window approach (using a 1-nucleotide step), recording the number of all variants and common variants, irrespective of their predicted effect, within each window, to eventually calculate a single-nucleotide resolution genome-wide intolerance score (Fig. 1).

The gwRVIS method is based on two hyperparameters: the length of the window and the minor allele frequency (MAF) threshold over which we consider a variant as common. We have fine-tuned these parameters by testing a range of values (Supplementary Fig. 1) and identified low sensitivity around the exact value selection. Taking into consideration the largest segregation achieved between the most intolerant and tolerant genomic classes (see "Methods") we selected a window length of 3 kb and an MAF threshold of 0.1%. As the number and ancestral diversity of sequenced human genomes increases, we expect this will permit smaller window sizes, which would provide better resolution for identifying smaller noncoding intolerant regions of the human genome. At the same time, there's been demonstrated value from using longer windows too, as previous works have shown that genetic contribution of variants relies on the genomic context of broader regions[14], including GC content, which has been correlated with various features of genome organization such as mutation rate and distribution of various classes of repeated elements[15].

In constructing the scores, we also perform a variant pre-processing quality control step, accounting for genomic coverage, variant-calling confidence, and simple repeat regions (Fig. 1; see "Methods"). We then fit an ordinary linear regression model to predict common variants based on the total number of all variants found in each window. As with the original RVIS formulation, we define the studentized residuals of this regression model as gwRVIS, with lower gwRVIS values corresponding to greater intolerance (Fig. 2a). Notably, a set of highly tolerant windows emerges, particularly enriched for chromosome 6 and specifically for regions of the human leukocyte antigen complex (Fig. 2b), consistent with a previously reported high degree of positive selection in this region of the genome[16].

**Stratifying the human genome based on intolerance to variation**. To determine biological relevance of gwRVIS we first sought to confirm the ability of gwRVIS to differentiate between different classes of protein-coding consensus coding sequence (CCDS) windows based on their disease relevance. We looked at the gwRVIS distributions across four sets of CCDS windows: OMIM-Haploinsufficient, 25% most intolerant (as defined by RVIS), 25% most tolerant (as defined by RVIS), and the remainder of CCDS (Fig. 2c). Despite not incorporating functional protein-coding annotations in constructing gwRVIS, we observe that it still manages to correctly stratify the four CCDS sets based on their expected levels of constraint, grouping them in order of decreasing intolerance to variation as follows: OMIM-Haploinsufficient, 25% most intolerant CCDS, rest of CCDS and 25% most tolerant CCDS (Fig. 2c). Any two adjacent CCDS sets in this ranking have returned genome-wide statistically significant difference (Mann–Whitney $U$; $p < 5 \times 10^{-8}$), except for the OMIM-Haploinsufficient and 25% most intolerant CCDS sets with a non genome-wide significant divergence (Mann–Whitney $U$; $p = 6.8 \times 10^{-5}$), which is expected as they both comprise of highly intolerant subsets of the human exome.

Having shown the ability of gwRVIS to successfully predict human disease based on greater intolerance detected among the protein-coding CCDS windows overlapping human disease genes, we next compared the intolerance of different regulatory genomic

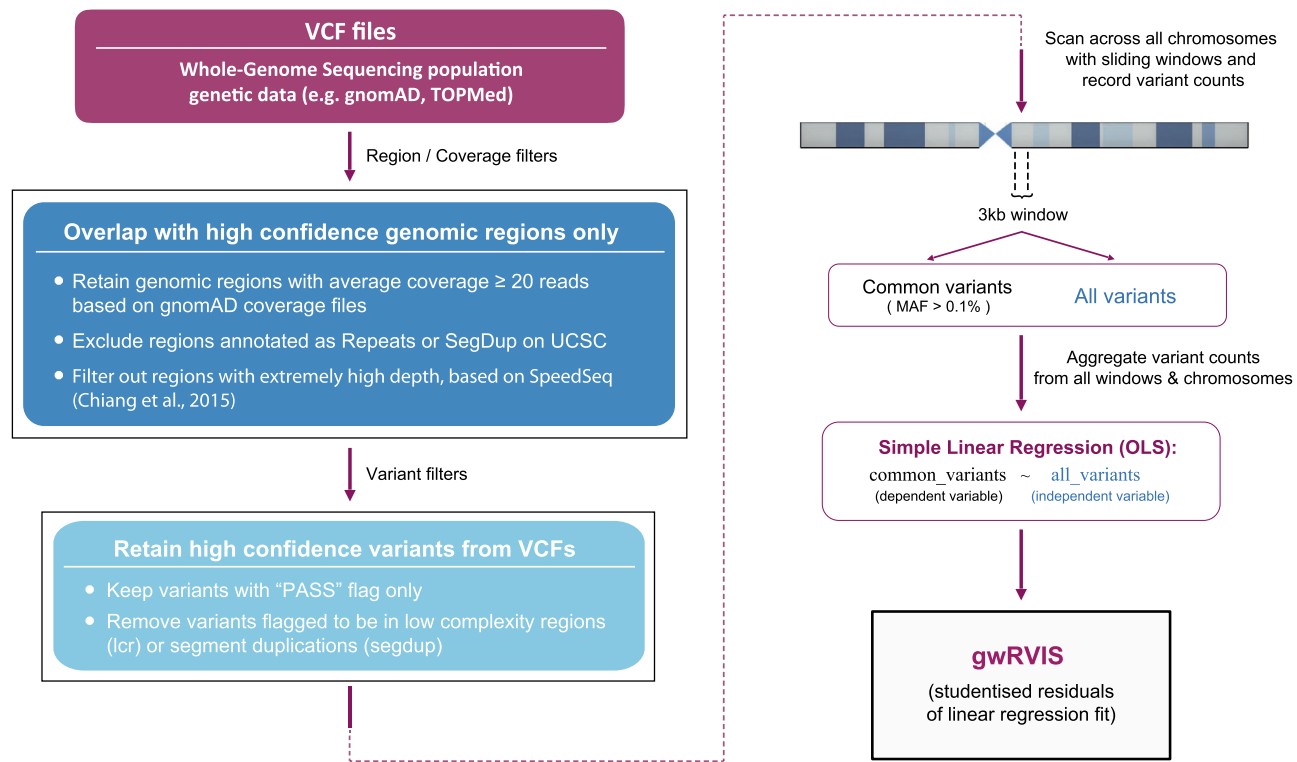

**Fig. 1 Genome-wide residual variation intolerance score (gwRVIS).** Flowchart for calculation of single-nucleotide resolution gwRVIS from whole-genome sequencing population genetic data (e.g., TOPMed and gnomAD), including relevant filtering steps (regarding depth coverage, repeat sequences, and variant quality).

classes, as captured by gwRVIS. We studied seven major genomic classes: intergenic regions, lincRNAs, introns, CCDS, UTRs, VISTA enhancers[6], and ultraconserved noncoding elements or UCNEs (see "Methods"), listed here in ascending order of inferred intolerance to variation (Fig. 2d). The intergenic gwRVIS score distribution emerges as the most tolerant class with a median gwRVIS = −0.0014. This median gwRVIS closely aligns with the theoretical null distribution defined by gwRVIS = 0, reflecting an equal number of observed and expected common variants. This validates the instinctive use of intergenic gwRVIS distribution as an empirical null distribution throughout the rest of the paper.

Surprisingly, the CCDS protein-coding region of the genome was not the most intolerant functional class. We observed that UCNEs[17] (highly conserved noncoding regions between human and chicken) are ranked as the most intra-species intolerant class (median gwRVIS: −0.99; Mann–Whitney $U$ vs. intergenic: $p < 1 \times 10^{-308}$), and this is despite gwRVIS not using any information about species conservation in its construction (Fig. 2d). This is consistent with previous observations that UCNEs are depleted of common variants in humans[18]. VISTA enhancers (a class of highly conserved enhancers active during embryonic development) and CCDS follow with the next highest intolerance to variation profile, very similar to UTRs (median gwRVIS: −0.77, −0.55, and −0.51; Mann–Whitney $U$ vs. intergenic: $p < 1 \times 10^{-308}$ for VISTA enhancers, CCDS and UTRs, respectively). Finally, introns and lincRNAs have a more tolerant gwRVIS score distribution that more closely resembles the distribution from intergenic regions, but due to the sheer size of the corresponding score lists the divergence remains highly significant (median scores: −0.050, −0.0015; Mann–Whitney $U$ vs. intergenic: $p < 1 \times 10^{-308}$ and $p = 2.6 \times 10^{-168}$, respectively).

An important aspect to note is that these comparisons currently reflect the class distribution effect of the seven mutually exclusive classes studied. It is evident that within each class there

are more and less tolerant windows in the human genome (Fig. 2d and Supplementary Fig. 2a). Thus, even among the intron and intergenic windows, there are some windows that achieve lower gwRVIS estimates than other windows overlapping with, for example, the CCDS sequence, and herein lies the true potential of this score to facilitate identification of more critical regions across all annotated regulatory classes.

We then test the predictive power of gwRVIS score distribution to distinguish windows overlapping the most intolerant (UCNEs) and tolerant (intergenic) genomic classes. We fit a logistic regression model with fivefold cross-validation to classify UCNEs- and intergenic-derived genomic windows, solely based on the gwRVIS score and find that gwRVIS achieves a notable area under curve (AUC) performance of 0.81 (standard deviation: 0.05) in distinguishing UCNE overlapping windows from all intergenic windows (Supplementary Fig. 2b). We also fit the same model but focused only on windows under negative selection (gwRVIS < 0), as detection of positive selection has been proven challenging[19]. In this case, gwRVIS achieves an even better performance with an AUC score of 0.86 (standard deviation: 0.02; Supplementary Fig. 2c).

**gwRVIS predictive power for functionally important genomic elements.** As we inspect the gwRVIS distributions across the different genomic classes, we observe a large variance in intolerance to variation within each class (Supplementary Fig. 2a). Thus, beyond the insight about which individual genomic classes are more/less intolerant relative to other classes, the gwRVIS distribution within a class also provides an extra dimension when trying to prioritize elements with the same functional annotation in terms of their pathogenicity potential or biological importance.

We earlier showed this to be true within the CCDS genomic class, in which known human disease genes had significantly

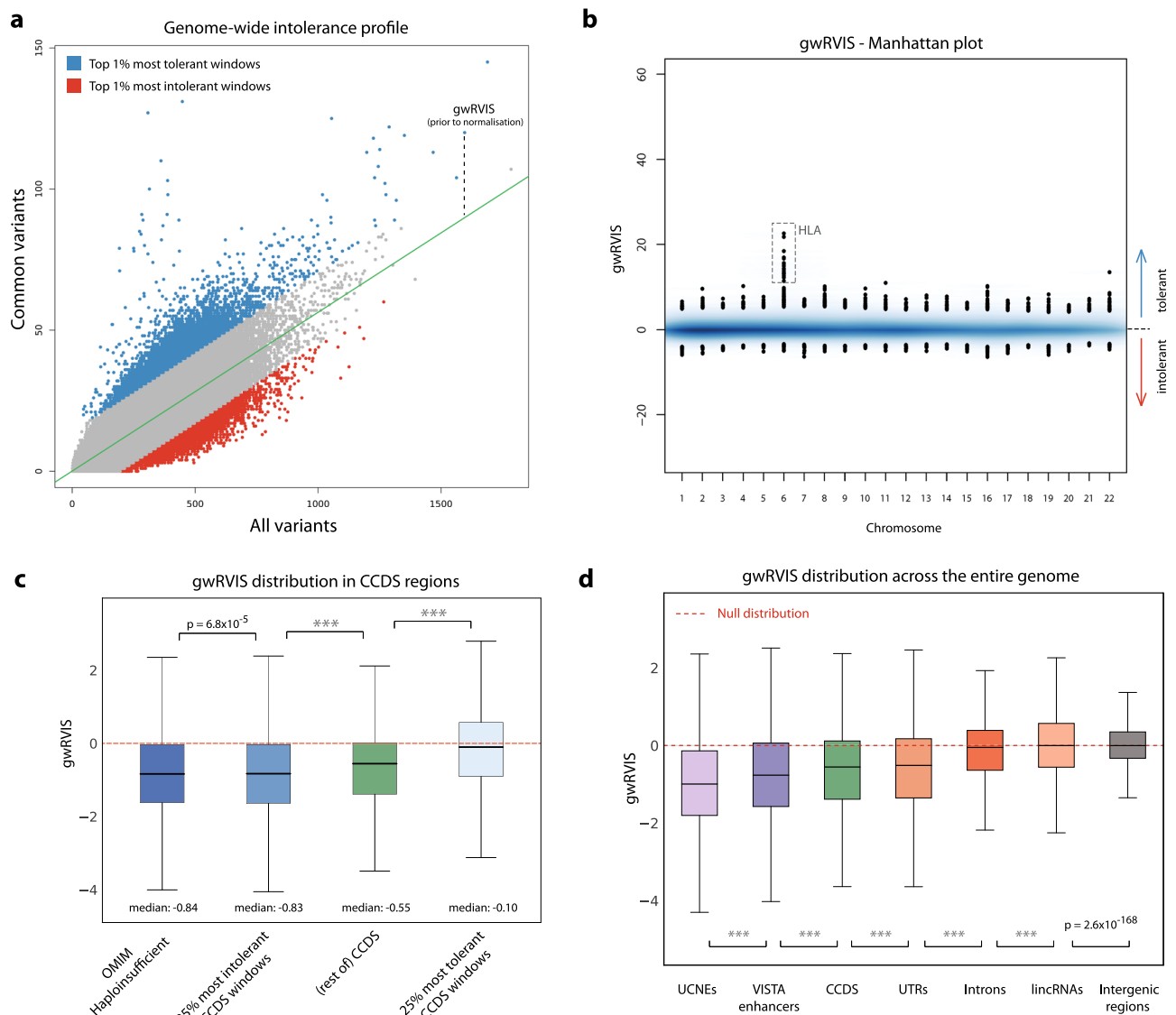

**Fig. 2 Genome-wide profile and performance of gwRVIS. a** Intolerance to variation profile across all genomic windows (having 3 kb length). A regression line (shown in green) is fit between "all" and "common" (MAF > 0.1%) variants across all windows. gwRVIS can be visualized as the vertical distance of each data point from the regression line (prior to normalization by the standard deviation of the total distribution). Red dots represent the top 1% of most intolerant windows (i.e., having fewer common variants than expected) while blue dots represent the top 1% of most tolerant windows. **b** gwRVIS distribution across all chromosomes, as extracted from the TOPMed dataset. A highly tolerant set of windows in chromosome 6 is enriched for HLA complex regions. **c** gwRVIS scores distribution (with single-nucleotide resolution) across different sets of mutually exclusive CCDS windows: OMIM-Haploinsufficient, 25% most intolerant (based on RVIS), 25% most tolerant, and rest of CCDS. *P* values from two-sided Mann–Whitney *U* tests are also provided for each pair of "adjacent" coding region classes in order of increasing intolerance. **d** Distribution of gwRVIS scores across different coding and noncoding genomic classes, in descending order of intolerance to variation: UCNEs, VISTA enhancers, UTRs, CCDS, introns, lincRNAs, and intergenic regions. The red horizontal dashed line (gwRVIS = 0) represents the mean of the theoretical null distribution (i.e., where the observed number of common variants equals the expected number). Intergenic regions are normally distributed around the null distribution, which validates their use as an empirical null distribution. Two-sided Mann–Whitney *U* has been employed to compare the gwRVIS distributions across all pairs of genomic classes (***$p < 1 \times 10^{-308}$). For each boxplot, its central line represents the median, the bounds represent the 25th and 75th percentile, and the whiskers extend up to 1.5 the interquartile range from the respective bounds.

lower gwRVIS than human CCDS genes not linked to human disease. To assess this signal outside of the protein-coding sequence, we first looked at the FANTOM5[20] gwRVIS scores to examine where VISTA enhancers[21] preferentially fall within that distribution. FANTOM5 is a resource that contains mammalian promoters, enhancers, lincRNAs, and miRNAs, including a collection of nearly 20,000 functional lincRNAs in humans. VISTA enhancers are important as they represent experimentally validated human and mouse noncoding

fragments with gene enhancer activity. We found, that among the 10% most intolerant FANTOM5-overlapping windows there was significant enrichment for VISTA enhancers compared to the 10% most tolerant end (approx. 7.5% vs. 1.5%, i.e., 5-fold enrichment; Fisher's exact test *p* value = $1.85 \times 10^{-62}$; Supplementary Fig. 3). This result demonstrates that the intolerant tail of the gwRVIS distribution for enhancers is more likely to be enriched for genomic elements of increased functional significance.

We have also compared gwRVIS to Orion[9], another method that relies on human genomic constraint, and found that gwRVIS can better predict CCDS vs. non-CCDS regions (Mann–Whitney $p$ value $= 8.4 \times 10^{-15}$ compared to 0.001 when using Orion; Supplementary Fig. 4; see "Methods").

Finally, in order to evaluate the sensitivity on different WGS datasets used as input for the intolerance score construction, we also calculated gwRVIS based on the gnomAD dataset ($n = 15,708$ individuals) and found it to be significantly correlated with the scores constructed using the TOPMed dataset (Pearson's $r$: 0.91; $p$ value $< 2.2 \times 10^{-16}$; Supplementary Fig. 5; see "Methods"), indicating the robustness of the method on comparable WGS datasets.

**Classification of noncoding pathogenic variants based on their intolerance to variation.** Overall, noncoding variants represent a small fraction of all pathogenic classified variants residing among curated variant-level resources such as ClinVar[22]. Here, we examine the properties of gwRVIS in the context of ClinVar clinically classified pathogenic noncoding variants. We compiled two lists of noncoding variants: a pathogenic set based on ClinVar and a set of benign variants based on the "control" variants from denovo-db[23] (see "Methods"). We have intentionally avoided constructing a negative set of benign variants based on common variants found in a large population cohort (e.g., gnomAD or TOPMED) due to the fact that gwRVIS construction is inherently informed by common variants distribution and, thus, this would introduce circularity in the model's prediction performance.

Across the six non-CCDS genomic classes (Fig. 2d), we identified the following numbers of noncoding pathogenic variants (Supplementary Fig. 6): 427 in UTRs, 47 in intergenic regions, 47 in lincRNAs, 2 in VISTA enhancers, 0 in UCNEs, and 5052 in introns (across 125, 22, 6, 2, 0 and 1606 unique 3 kb windows, respectively). We also captured 51,230 variants across 3618 unique windows for CCDS regions based on the ClinVar annotation.

We then trained a logistic regression model with fivefold cross-validation, using gwRVIS or another genome-wide score (CADD[24], phastCons46way[25], phyloP46way[26], and Orion[9]) as the only independent variable predictor (see "Methods"). We focused on the four noncoding genetic classes that have at least 20 distinct noncoding windows harboring pathogenic variants (lincRNAs, UTRs, intergenic, and introns). Remarkably, we observed that gwRVIS achieves the highest performance in pathogenic variant classification from lincRNA regions (AUC = 0.937; Fig. 3a), which is significantly better than the performance of phyloP46way, phastCons46way and Orion but not from CADD (DeLong test: $p = 5.83 \times 10^{-3}$; $p = 1.7 \times 10^{-3}$ $p = 0.012$ and $p = 0.64$, respectively; Supplementary Fig. 16 and Supplementary Table 1). Similarly, gwRVIS performs better than all other scores in intergenic regions (AUC = 0.763; Fig. 3b), with significantly better performance than phyloP46way, phastCons46way and Orion but not CADD (DeLong test: $p = 2.65 \times 10^{-6}$; $p = 6.2 \times 10^{-3}$ $p = 0.02$ and $p = 0.34$, respectively; Supplementary Fig. 16 and Supplementary Table 2). In UTRs, gwRVIS's performance drops but remains significantly higher than Orion (AUCs: 0.732 vs. 0.597, DeLong test $p = 4.71 \times 10^{-4}$; Fig. 3c and Supplementary Table 3).

In order to estimate the contribution of gwRVIS information in noncoding variant detection, we also trained a multiple logistic regression model using gwRVIS and CADD as the independent variables. We observe, that gwRVIS significantly boosts CADD's performance (Fig. 3) in lincRNAs (AUC: increased to 0.937 from 0.895; DeLong test $p = 0.036$), intergenic regions (AUC: increased to 0.809 from 0.741, even though non significantly; DeLong test $p = 0.07$) and UTRs (AUC: increased to 0.835 from 0.777; DeLong

test $p = 4.18 \times 10^{-23}$). This indicates that gwRVIS captures orthogonal information that is not represented among the 63 distinct annotations/features employed by CADD. Moreover, although ncRVIS is the top-performing single score in UTRs (AUC = 0.823), it is lower than the combined gwRVIS and CADD score (AUC = 0.835).

We have also observed that gwRVIS is not superior in classifying pathogenic variants within protein-coding CCDS or intronic regions (Supplementary Fig. 7). This is expected as the intention of gwRVIS is to support the interpretation of the noncoding sequence where, unlike the protein-coding sequence, there is no information about variant effects. Overall, based on its optimal performance in non-coding regions using a cross-validation framework and the performance boost in the combined model with CADD, gwRVIS values are likely to be highly generalizable to other datasets when seeking to prioritize candidate variants in the non-coding genome.

**A multi-module deep learning framework for non-coding variant pathogenicity inference.** Equipped with a human-lineage-specific constraint score that spans the entire human genome we next sought to further improve our ability to prioritize noncoding sequence by integrating additional information beyond gwRVIS. To this end, we integrate two additional layers of information: (a) primary genomic sequence context around each variant (unstructured data) and (b) genomic annotations such as methylation, chromatin accessibility, or other structured features extracted from raw genomic sequences, such as GC content and mutability rate (see "Methods").

By combining this information (gwRVIS, primary genomic sequence context, and additional genomic annotations) we developed "Junk Annotation" RVIS or JARVIS: a multi-module deep learning framework for pathogenicity inference of noncoding regions that still remains naïve to existing phylogenetic conservation metrics in its score construction (Fig. 4a). We trained four different models for JARVIS: (a) Gradient Boosting using structured features (i.e., without raw sequence information), (b) feed-forward deep neural net (DNN) using structured features, (c) convolutional neural net (CNN) with raw sequence input, and (d) the multi-module neural network model that combines information from both structured features and raw sequences. The multi-module model outperformed all other models used for training JARVIS on the ClinVar pathogenic variant set (Fig. 4b and Supplementary Fig. 8), achieving an AUC of 0.940 with fivefold cross-validation (compared to AUC scores of 0.930, 0.929, and 0.872, from the DNN, Gradient Boosting, and CNN models, respectively). Thus, we define as JARVIS the scores extracted by the multi-module model, which comprises of a CNN module for information inference from underlying sequence and a feed-forward DNN to assess structured feature data such as gwRVIS, sequence-derived features, and external annotations (see also "Methods").

As with gwRVIS, our training set adopted all non-coding variants annotated in ClinVar as "Pathogenic" or "Likely pathogenic" and a random subset of "control" variants from denovo-db[23], considered to be benign (see "Methods"). To build a generic noncoding variant classification model, we integrated variant data from five noncoding regions during training: intergenic regions, UTRs, lincRNAs, UCNEs, and VISTA enhancers. We did not include introns given the low empirical performance of gwRVIS in these regions.

We compare JARVIS against ten other popular genome-wide scores (Fig. 4b and Supplementary Fig. 9): Orion[9], CADD[24], phastCons[25] (46way), phylop[26] (46way), DeepSEA[27], ncER[28], LINSIGHT[29], Eigen-PC[30], DANN[31], and CDTS[10]. It is important

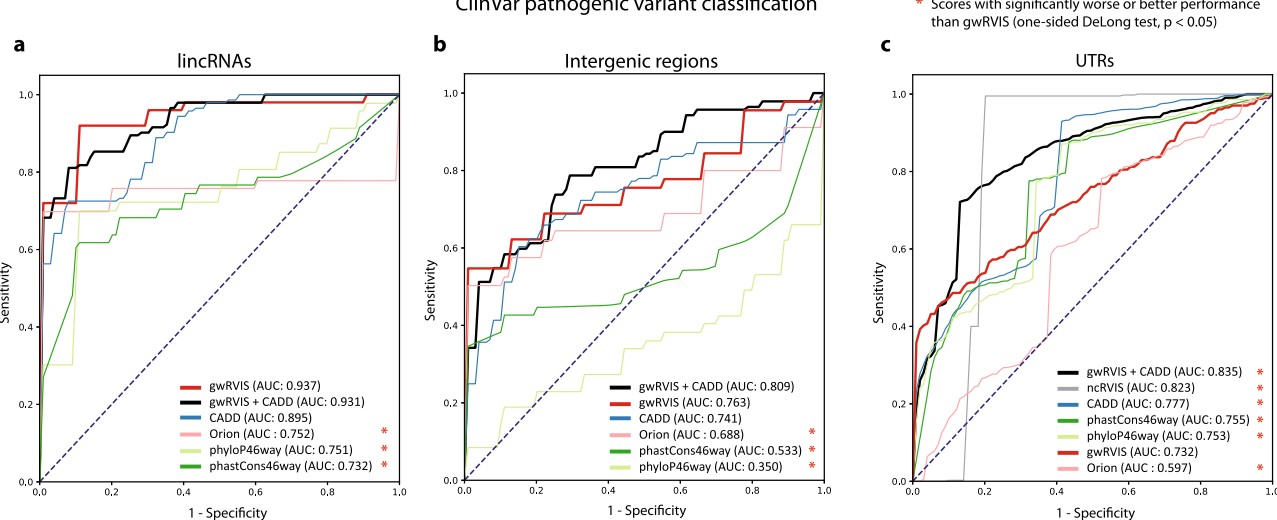

**Fig. 3 Predictive power of gwRVIS for pathogenic variant classification.** Mean ROC curves (with fivefold cross-validation) from gwRVIS benchmarking against CADD, phastCons (46-way), phyloP (46 way), and Orion, during ClinVar-pathogenic vs. denovodb-benign variant classification for three noncoding genomic classes: **a** lincRNAs, **b** intergenic regions, and **c** UTRs. The combined performance of gwRVIS with CADD is also shown. ncRVIS is included in the benchmarking of the UTR regions (**c**), as a robust score specifically designed for the UTR genomic class. One-sided DeLong's tests have been performed to assess the statistical significance of the differences in predictive power between gwRVIS and the rest of genome-wide scores.

to note that ncER, LINSIGHT, CADD, and DANN incorporate multiple phylogenetic conservation metrics (e.g., phyloP, phast-Cons, SiPhy, and CEGA) in their score constructions. It is expected that a large proportion of ClinVar variants are annotated as pathogenic based on occurring at evolutionarily conserved sequences[22] among other evidence. This can introduce circularity when using conservation-based metrics as a predictor of literature-based pathogenic noncoding variants. In addition, these four scores have been directly or indirectly trained with a subset of the JARVIS training set. Specifically, ncER has been trained based on ClinVar (and HGMD) pathogenic variants, which comprise a subset of the pathogenic set used by JARVIS and thus its performance on the JARVIS training set may suffer from data leakage. In addition, LINSIGHT, ncER, CADD, and DANN have been trained with prior knowledge of proximity to CCDS (distance to TSS) and annotation of a region as a predicted distal regulatory module. A large proportion (>55%[29]) of ClinVar noncoding pathogenic variants are proximal to coding regions (classified as splicing or UTR variants) while a large part of the rest has been annotated based on a known regulatory effect[22]. Thus, ncER, LINSIGHT, CADD, and DANN could be impacted by ascertainment bias due to the variant distribution of the JARVIS training set with respect to closest TSS, however, we still report their performance for reference.

We trained the deep learning-based multi-module JARVIS model using all ClinVar noncoding pathogenic variants across all chromosomes with fivefold cross-validation (randomized splits) and compared its performance against the rest of the scores (see "Methods"). JARVIS performs significantly better than all other scores (AUC = 0.940; Fig. 4b; DeLong test $p < 2.710^{-4}$; $p$ values from each test available in Supplementary Fig. 19a and Supplementary Table 4), except for DeepSEA (AUC = 0.945; Fig. 4b; DeLong test: nonsignificant $p = 0.064$), despite some of them including conservation information. Two scores, ncER and LINSIGHT, achieved better performance on this dataset (AUC: 0.977 and 0.961; DeLong test: $p = 2 \times 10^{-5}$; $p = 0.024$, respectively; Supplementary Fig. 19a and Supplementary Table 4). However, ncER's predictions are likely boosted by data leakage of the current JARVIS training set in its own training set and both scores may be biased with additional information, such as

distance from the closest TSS. When integrating the TSS distance in the JARVIS model, this version of JARVIS indeed exceeds the performance of both LINSIGHT and ncER (AUC = 0.984; Supplementary Fig. 9). However, we do not eventually include TSS distance as a feature in the final model as we want to avoid biasing JARVIS predictions toward variants residing closer to protein-coding regions.

The fully randomized cross-validation strategy for assessing JARVIS training performance may be prone to overestimation of its generalizable performance due to the annotated ClinVar pathogenic noncoding variants being preferentially closer to protein-coding genes compared to a random set of control variants derived from denovo-db. We have thus prepared an alternative version of the training set (matched training set) by selecting control variants from denovo-db with very similar distribution of TSS distances to closest genes, as compared to the pathogenic variants employed in the JARVIS training set (Supplementary Fig. 17). JARVIS performs significantly better than all other scores (AUC = 0.800; Supplementary Fig. 18; DeLong test $p < 0.0037$, Supplementary Fig. 19b and Supplementary Table 5) except for the only ncER, which has significantly better performance (AUC = 870; DeLong test $p = 4.93 \times 10^{-5}$) but likely boosted by data leakage.

We also employed an alternative cross-validation strategy by stratifying variants based on their chromosome location. This strategy ensures that variants from the same genomic region cannot be part of both the training and test sets at any cross-validation step, thus removing any bias from data circularity. JARVIS performance dropped marginally (AUC = 0.929, Supplementary Fig. 18) but remained significantly higher than all others (DeLong test $p < 5.510^{-11}$; Supplementary Fig. 19c and Supplementary Table 6) except for DeepSEA and CADD that performed nonsignificantly worse (AUC = 0.922 and 0.913, DeLong test $p = 0.065$ and 0.158, respectively) and ncER and LINSIGHT that performed significantly better (AUC = 0.980 and 0.969, DeLong test $p = 1.16 \times 10^{-8}$ and $2 \times 10^{-4}$, respectively).

Based on the performance of JARVIS when using different models for training (Fig. 4b and Supplementary Fig. 8), we observe that deep learning models are superior to Gradient Boosting and also that the inclusion of raw sequences features

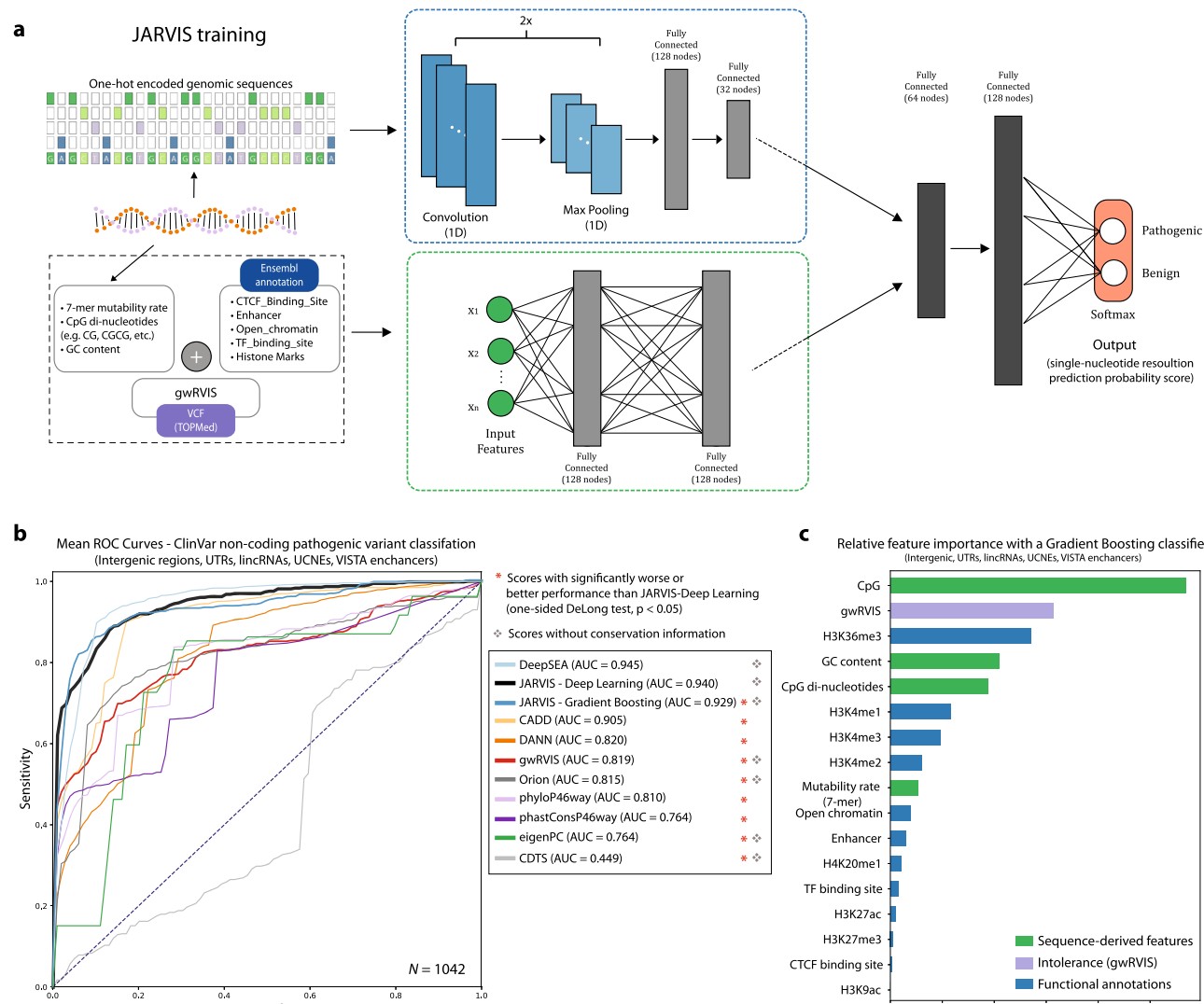

**Fig. 4 JARVIS: a multi-module deep-learning-based score for non-coding variants pathogenicity inference with single-nucleotide resolution. a** Deep-learning framework for noncoding variants pathogenicity inference based on different types of annotation: genome-wide residual variation intolerance score (gwRVIS), primary genomic sequence, structured features extracted by raw genomic sequence (mutability rate, GC content, etc.) and additional annotations from Ensembl (histone marks, chromatin accessibility, CTCF-binding sites, etc.). All structured features are initially passed onto a two-layer deep neural net (DNN). Primary genomic sequences (in windows of 3 kb) are fed into a deep convolutional neural net and then flattened prior to merging with the higher representations of the structured features previously processed by the DNN. The combined higher representations of features are processed by two additional fully connected layers, followed by a "softmax" output, which gives the pathogenicity likelihood for each input variant as a probability score. **b** JARVIS performance with fivefold cross-validation after training with a multi-module deep neural network, using all noncoding ClinVar-based pathogenic variants and a matched set of putative benign variants from denovo-db. Variants used for training belong to any of the following genomic classes: intergenic regions, UTRs, lincRNAs, UCNEs, or VISTA enhancers. A total of 521 noncoding pathogenic variants have been used for this classification task, thus $N = 1042$ represents the total size of the training set (using a set of control variants of equal size). Performance for the rest of the genome-wide scores shown here has been calculated using a logistic regression model with fivefold cross-validation on the JARVIS training set. **c** An overview of the relative importance of the structured features integrated within JARVIS, as they are extracted by a Gradient Boosting classifier following an impurity-based feature selection algorithm.

provides the highest predictive power in the pathogenic variant classification task from non-coding regions. In order to assess the relative contribution of the structured features, we initially performed a correlation analysis (based on Pearson's $r$ coefficients) between all pairs of structured features and observed that gwRVIS has little correlation with the rest, highlighting that it represents an orthogonal type of information not routinely captured by the rest of the JARVIS features (Supplementary Fig. 10). The sequence-derived features (GC content, mutability rate, etc.) and some of the histone marks were found to be highly

correlated with each other within the respective feature group (Supplementary Fig. 10).

It is, however, difficult to infer the real contribution from each feature employed by the full deep learning-based JARVIS model. Thus, we employ an impurity-based feature extraction algorithm with a Gradient Boosting classifier as a proxy to infer the relative contribution of each of the structured features. We observe that gwRVIS ranks second in feature importance while the sequence-derived features, specifically CpG density, are the other most dominant subset from the entire feature set (Fig. 4c). Notably,

CpG density in promoter regions has been associated with loss-of-function intolerance of proximal coding regions[32]. Thus, JARVIS may preferentially prioritize noncoding variants that have a traceable functional effect on coding regions. Functional annotations follow with lower contribution but still carrying a considerable burden, especially for certain types of histone marks. These findings demonstrate that the genome-wide intolerance score, as it is captured by gwRVIS, adds considerable value to the predictive power of JARVIS.

We also sought to explore the feature contribution from the CNN module of the JARVIS framework and assess whether it has learnt any biologically meaningful motifs. For this analysis, we relied on previous approaches that look into the most highly activated filters that comprise the feature maps at the output of the first convolutional layer[33]. Specifically, we selected the most activated $k$-mers (where $k = 11$, equal to the selected filter size) in the training set of pathogenic input sequences and aligned them with known motifs (see Methods). We observed that, despite the fairly small size of the training set, JARVIS CNN module has managed to learn dozens of sequence patterns that align with known vertebrate, human or mouse motifs, many of which are enriched for long Cytosine stretches and thus GC base pairs too (Supplementary Figs. 22, 23, 24 and Supplementary Data 1).

**JARVIS model validation on independent datasets**. To further assess the generalized performance of JARVIS, we screened throughout the literature to identify additional independent sets of noncoding variants, either annotated as pathogenic or of high functional importance. For this task, we employed four different sets of noncoding variants described in Wells et al.[28] (Fig. 5): (a) a set of 631 genome-wide association studies (GWAS) hit single-nucleotide variants (SNVs), (b) a set of 54 noncoding variants associated with Mendelian traits, and (c, d) two "generalization" test sets from noncoding RNAs or other regions[28], with 35 and 17 hits, respectively. In all cases, we made sure to remove from each independent validation set any variants that were included in the JARVIS training set to avoid over-optimistic predictions induced by data leakage.

JARVIS significantly outperforms all scores in classifying GWAS hit SNVs (AUC = 0.760 vs. 0.661 from the second-ranked LINSIGHT; Fig. 5a—DeLong test $p < 7.65 \times 10^{-7}$; Supplementary Fig. 20 and Supplementary Table 7). It also performs significantly better than all other scores in the prediction of Mendelian noncoding variants (AUC = 0.988, DeLong test $p < 6.70 \times 10^{-3}$; Supplementary Fig. 20 and Supplementary Table 8) except for ncER which achieves a nonsignificantly higher performance (AUC = 0.990; Fig. 5b). In the third validation set (ncER-derived "ncRNA" generalization dataset) JARVIS ranks fourth (AUC: 0.956, Fig. 5c), significantly outperforming Eigen-PC, DANN, Orion, phastCons, and CDTS (DeLong test $p < 5.18 \times 10^{-3}$, Supplementary Fig. 20 and Supplementary Table 9), with LINSIGHT, ncER, and DeepSEA performing nonsignificantly better (AUC = 0.991, 0.963, 0.962; DeLong test $p = 0.34$, 0.36, and 0.44, respectively; Supplementary Fig. 20 and Supplementary Table 9). Similarly, in the fourth validation set (ncER-derived "other" generalization dataset) JARVIS ranks fourth (AUC: 0.914, Fig. 5d), significantly outperforming phyloP, phastCons, DANN, Eigen-PC, Orion, and CDTS (DeLong test $p < 1.66 \times 10^{-3}$, Supplementary Fig. 20 and Supplementary Table 10), with LINSIGHT, ncER, and DeepSEA performing again nonsignificantly better (AUC = 0.988, 0.969, 0.938; DeLong test p = 0.32, 0.59, and 0.84, respectively; Supplementary Fig. 20 and Supplementary Table 10).

Notably, JARVIS performance in the latter two test sets was even lower when TSS distance was included as part of the JARVIS training feature set (AUC: 0.874 and 0.903, respectively; Supplementary Fig. 14).

We also tested the relative performance of the four JARVIS model variations (i.e., based on either Gradient Boosting or the deep-learning-based ones, including raw sequences or without). Deep learning-based models once again outperform the Gradient Boosting classifier. Moreover, we observe that the multi-module model, integrating both raw sequence information and structured data, outperforms the rest in two of the four validation sets (Supplementary Fig. 11). The deep learning model based solely on raw sequence information is the top performer in the other two sets and especially yielding a considerably higher performance on the GWAS validation set (AUC = 0.790). These results demonstrate the added value from leveraging raw sequence information with deep learning in such classification tasks.

Overall, despite its construction being restricted to human lineage and sequence context information, JARVIS either outperforms or performs comparably with state-of-the-art scores that integrate multiple conservation-informed metrics.

**Prioritization of putative pathogenic structural variants**. Structural variants are large DNA rearrangements[34] that have been implicated to have a profound impact in evolution and human disease[35–37]. We sought to estimate the ability of JARVIS and gwRVIS to distinguish large structural variants based on their inferred clinical impacts. We employ for this task a rich set of structural variants (SV) called from 14,891 whole genome sequences in the gnomAD dataset[35] (v2.1). Structural variants overlapping with protein-coding regions have been annotated with various functional consequences (Copy Gain, Duplication-LoF, Intronic, LoF, Partial-Duplication, Promoter, UTR, and Whole-Gene inversion) or are otherwise classified as intergenic SVs. We consider the latter case (SV in intergenic regions) as our negative set of non-dosage sensitive regions and try to classify it against all other genic structural variants, that have proven to be dosage sensitive. Structural variant lengths included in this dataset vary from two nucleotides to a few million (Supplementary Fig. 12). However, the largest proportion of variants from five out of the nine structural variant classes (intronic, UTR, intergenic, promoter, and partial duplication) have length distributions centered around 100–1000 base pairs. In order to classify pathogenic structural variants from benign ones, we represent each variant as a consensus of four summary statistics across its length. Specifically, we assign to each structural variant the average of the median, mean, first and third quartiles of the respective genome-wide score (see also Methods). Thus, each structural variant is eventually represented by a single value that captures the aggregate profile of a score's distribution within that region. We then used a 10-fold cross-validation approach based on a Logistic Regression model for benchmarking between the top-performing scores from the previous validation tests (JARVIS, LINSIGHT, ncER, CADD, Eigen-PC) along with gwRVIS and Orion.

We observe that JARVIS achieves the highest performance in six out of eight comparisons (Fig. 6a; AUC = 0.684–0.844), outperforming all other scores that follow with lower AUC ranges (Orion: 0.591–0.755; LINGISHT AUC: 0.605–0.747; ncER: 0.448–0.709, and gwRVIS: 0.542–0.667, ordered by the highest AUC in each range). In all these cases, JARVIS was significantly better than all other (DeLong test $p < 0.022$, Supplementary Fig. 21 and Supplementary Tables 11–16), except for LINSIGHT in the inversion-span Structural Variant test set, with a non-significantly better performance (AUC = 0.751 vs. 0.747, from JARVIS and LINSIGHT, respectively; DeLong test $p < 0.087$). As for the UTR-related Structural Variant class, JARVIS significantly outperforms ncER, gwRVIS, and Eigen-PC (AUC = 0.707; DeLong test $p < 6.12 \times 10^{-7}$) but is also significantly outperformed by LINSIGHT,

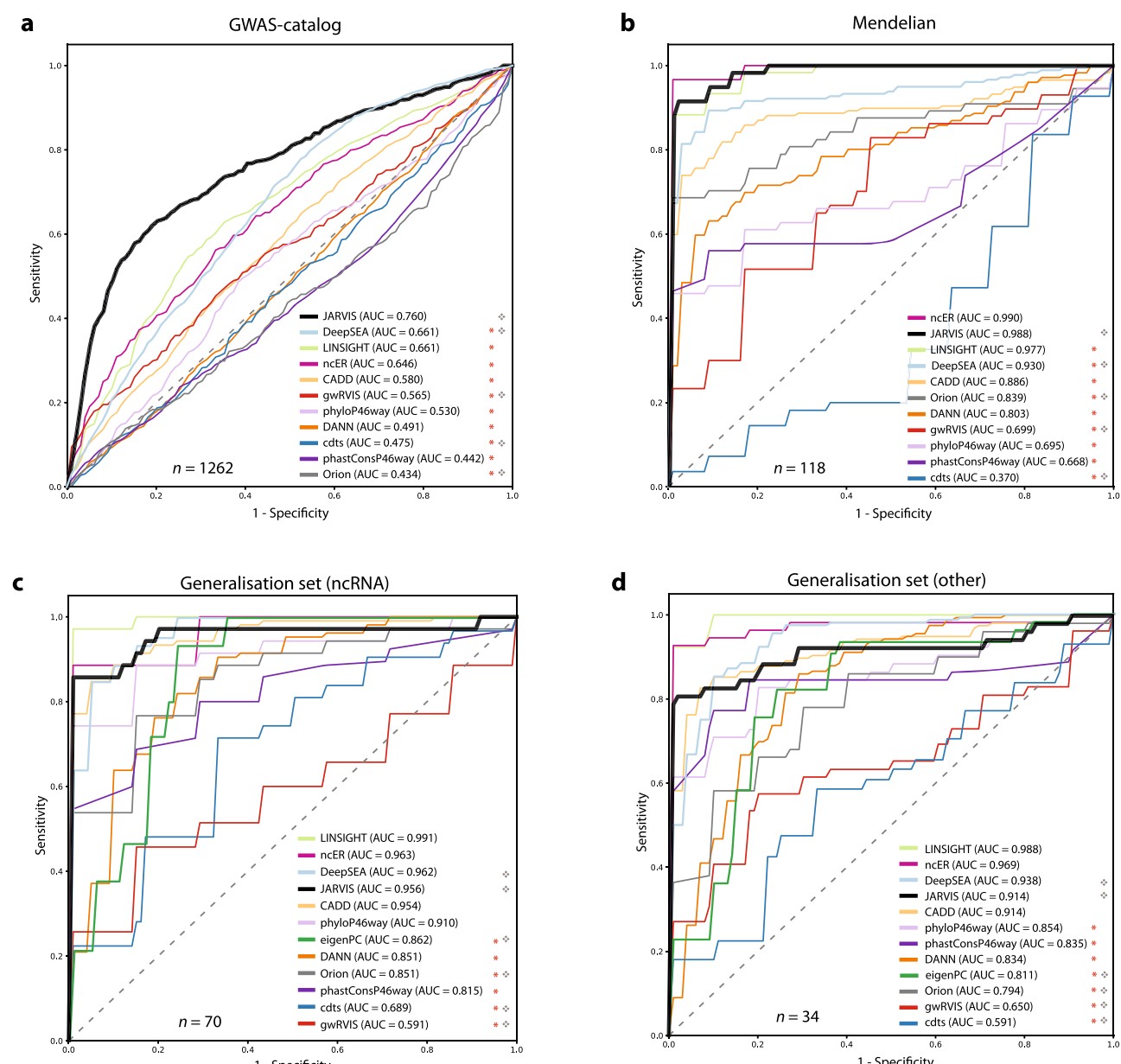

**Fig. 5 JARVIS performance on validation sets.** ROC curves from prediction on different sets of noncoding variants (falling into intergenic regions, UTRs, lincRNAs, UCNEs, and VISTA enhancers) not used during JARVIS training. In each case, a benign set of equal size has been randomly subset from the denovo-db control variants, avoiding any overlaps with the pathogenic variants in each of the validation sets. **a** GWAS hit SNVs (n = 1262). **b** Noncoding variants with mendelian traits (n = 118). **c** Generalization test set (ncRNA; n = 70). **d** Generalization test set (other; n = 34). In each plot, n refers to the total size of the validation set, including both pathogenic variants and a sample of control variants of equal size.

Orion, and CADD (AUC = 0.748, 0.747, 0.722, respectively; DeLong test $p < 0.023$; Supplementary Figs. 13a, 21 and Supplementary Tables 17 and 18). Finally, performance in Intronic regions is low across all scores, with Orion leading with an AUC of 0.587 (DeLong test $p = 9.86 \times 10^{-29}$) and gwRVIS and JARVIS closely following with AUC scores of 0.564 and 0.562, respectively (Supplementary Figs. 13a and 21).

Furthermore, we wanted to examine how structural variants called within a particular genomic class rank in terms of their intolerance to variation (as captured by gwRVIS) and

pathogenicity likelihood (as captured by JARVIS) compared to the rest of the genomic class. We expect that large genomic alterations observed among the general population should not have a detrimental effect on fitness and thus should map into more tolerant regions of the human genome. We focus on these types of comparisons on SVs called in UTRs and intronic regions, where a disease-relevant effect can be traced back to a specific protein-coding gene. We observe in UTRs that the called SVs have a significantly more tolerant profile (higher values) compared to the entire distribution (Fig. 6b; median gwRVIS: −0.020 vs. −0.51,

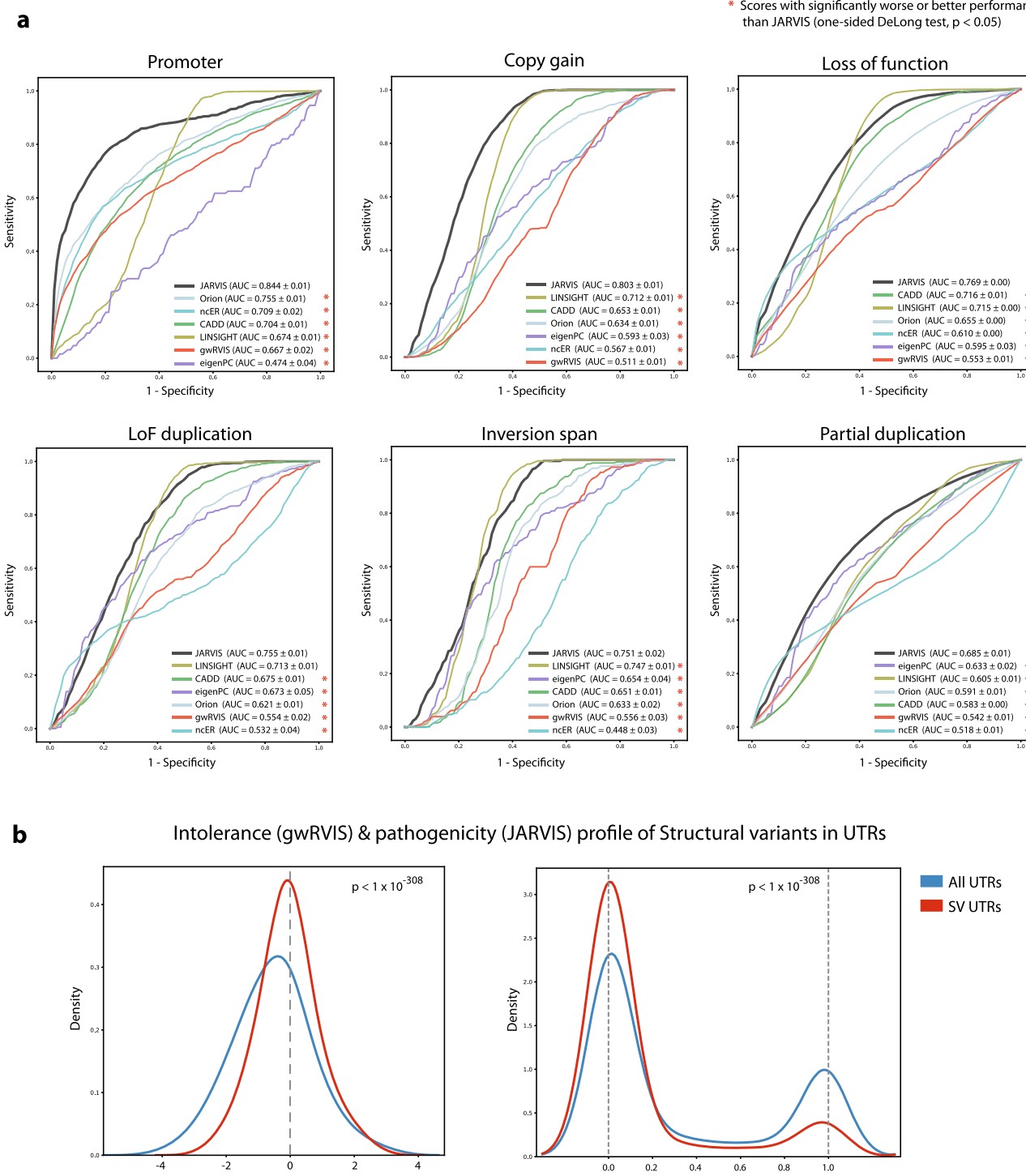

**Fig. 6 JARVIS and gwRVIS performance on structural variants. a** ROC curves from classification of nongenic structural variants (intergenic) against different sets of putative pathogenic genic SVs, using a tenfold cross-validation approach with a logistic regression model on five scores: JARVIS, gwRVIS, LINSIGHT, ncER, and Orion. One-sided DeLong's tests have been performed to assess the statistical significance of the differences in predictive power between JARVIS and the rest of genome-wide scores. **b** gwRVIS distribution across the UTR regions for the subset of called structural variants vs. the entire genomic class (median gwRVIS: $-0.020$ vs. $-0.51$, two-sided Mann–Whitney $U$ $p < 1 \times 10^{-308}$). Tolerance increases toward greater gwRVIS values. **c** JARVIS distribution across the UTR regions for the subset of called structural variants versus the entire genomic class (mean JARVIS: 0.147 vs. 0.326, two-sided Mann–Whitney $U$ $p < 1 \times 10^{-308}$). Pathogenicity likelihood increases toward greater JARVIS values.

Mann–Whitney $U$ $p < 1 \times 10^{-308}$) and the same pattern, albeit at a lower degree, emerges for intronic regions (Supplementary Fig. 13b; median gwRVIS: 0.051 vs. −0.039, Mann–Whitney $U$ $p < 1 \times 10^{-308}$), in accordance with our original hypothesis. Similarly, the pathogenicity likelihood of SVs in UTRs and intronic regions is lower compared to the rest of each distribution (Fig. 6b and Supplementary Fig. 13b) with respective mean JARVIS scores of 0.147 vs. 0.326 for UTRs and 0.119 vs. 0.142 for the Intronic regions (Mann–Whitney $U$ $p < 1 \times 10^{-308}$ in both). This result is again in accordance with the expectation of the lower pathogenic effect of large structural variations found in the general population.

## Discussion

Several methods have been developed in recent years that attempt to address the challenge of prioritizing noncoding variants. Most of these methods employ a combination of functional annotation but also cross-species conservation information. Here, we presented JARVIS and gwRVIS, two scores that encompass exclusively human lineage-specific information but still manage to perform comparably or even better than conservation-informed scores. The pathogenicity likelihood of noncoding regions cannot be efficiently inferred by cross-species conservation-based metrics due to the high evolutionary turnover of these regions. Thus, the two human-lineage-specific metrics we introduce may allow us to reduce dependence on conservation-derived metrics and increasingly rely on the human genomic constraints in our search for human disease variants.

One step toward improving the current JARVIS and gwRVIS implementation would be increasing the resolution achieved within genomic subregions. This can be done by employing larger WGS datasets, such as the prospective UK Biobank WGS dataset of 500,000 individuals, which would enable the selection of considerably shorter window sizes and a lower "common" MAF threshold. Moreover, we could explore expanding the functional annotation information with data from additional cell lines and emerging data from ENCODE 3[38]. Finally, increased inclusion of WGS representation from diverse ancestries will allow the construction of ethnicity-specific intolerance scores and enable a more refined prioritization of noncoding regions across individuals based on their genetic background.

## Methods

**Definition of high-confidence genomic regions and variant filtering.** In order to eliminate biases from extreme/low coverage genomic regions, we have defined a set of high-confidence regions (Fig. 1). Specifically, we first retained genomic regions with average coverage >20 reads, based on gnomAD coverage files (version 2). We then filtered out regions with exceptionally high depth, as captured by SpeedSeq[39]. Finally, we excluded regions annotated as Repeats or Segment Duplications (SegDup) at UCSC[40].

**Variant filtering from TOPMed and gnomAD.** VCF files from TOPMed (freeze5 release) and gnomAD (version 2.1.1) were overlapped with a custom set of high-confidence regions (described earlier) and their variants were further filtered, prior to gwRVIS calculation (Fig. 1). Specifically, only SNVs with a "PASS" annotation flag were retained. In addition, variants annotated with a low complexity region ("lcr") or segment duplication ("segdup") flag were removed. After all filtering steps, the resulting genomic area of higher confidence comprised 2.4 billion base pairs in total (i.e., about 75% of the genome), for which gwRVIS scores could be calculated.

gwRVIS's regression model and extracted scores were eventually constructed based on all 3 kb (nonoverlapping) windows that overlapped with the high-confidence regions and contained high-confidence variant data, as defined above. Regions that were "masked" during preprocessing were not considered during gwRVIS construction and have not been assigned a gwRVIS score.

**Mapping of regions and variants to a genomic class.** We have extracted the genomic coordinates for each genomic class from the Ensembl Human annotation GRCh37 (release 75). When assigning gwRVIS and JARVIS scores to genomic classes, we intersect each nucleotide position with the genomic region defined by

each class using bedtools v.2.29[41]. This is done in a hierarchical manner, with different priorities given to each genomic class, as a single nucleotide position may belong to more than one genomic classes. Specifically, genomic classes are processed in the following order of descending priority (with coding classes having a higher priority than non-coding in general): OMIM-Haploinsufficient genes, 25% RVIS-predicted most intolerant CCDS, 25% most tolerant CCDS, rest of CCDS, VISTA enhancers, miRNAs, UCNEs, UTRs, introns, lincRNAs and intergenic regions. Upon each intersection cycle, the genomic class region intersecting with the reference windows set is also subtracted by it to create an updated reference set of genomic regions, to be used by the next genomic class in priority. This ensures that genomic positions that belong to multiple classes are claimed only by a single class, the one with the highest priority among them.

When it comes to variant classification by type (pathogenic or benign), we follow a more stringent approach, built on top of the original genomic stratification described above, in order to assign each variant to a genomic class. We have used the ClinVar and denovo-db databases to extract known coding and non-coding pathogenic or benign variants, respectively, to assess the predictive power of gwRVIS, JARVIS, and other previously published genome-wide scores. denovo-db is a collection of germline de novo variants identified in the human genome and those annotated with a "control" phenotype are considered as not having a pathogenic effect. As gwRVIS is constructed with a window-based approach, we wanted to make sure that in case a window contained noncoding ClinVar-annotated variants it was not contaminated with a large amount of coding sequence. That aims to eliminate the risk of interpreting the contribution of a noncoding variant based on the presence of proximal coding variants or coding sequence in general. So, during gwRVIS construction, we dynamically compile a "blacklist" of genomic regions that contain both coding and non-coding variants and where the region covered by coding sequence is 10% more than the non-coding one. These "blacklisted" windows are then used to assess only coding variants during scores benchmarking and are excluded from non-coding variants classification (e.g., from UTRs and intergenic regions). Eventually, 4 and 81 variants from ClinVar were excluded from the intergenic and UTR regions, respectively, that contain both coding and noncoding variants, so that pathogenicity inference of noncoding variants is not contaminated by any large contribution from the coding sequence.

**Window length and MAF hyperparameter tuning for gwRVIS construction.** The window length and MAF hyperparameters have been optimized based on the best segregation achieved between non-intergenic and intergenic regions, as measured by the AUC score of the logistic regression fit for the binary classification using gwRVIS (Supplementary Fig. 1a). In order to reduce the parameter search space, we performed a two-step optimization approach instead of a full-scale grid search. We initially selected a MAF value of 0.1% as a nominal threshold for common variant annotation, based on previous studies analyzing cohorts of similar size[42], aiming to first optimize the window length. We employed 11 window values for the window length sensitivity analysis: 500, 1k, 2k, 3k, …, 10k bp, and used the center from all genome-wide tiled (nonoverlapping) windows as the representative value for the respective window, to perform the sensitivity analysis. UCNEs consistently achieved the best segregation against intergenic regions, with higher AUC scores achieved in all cases compared to the rest of examined genomic classes. We observed that AUC scores keep increasing for the majority of genomic classes as the size of the window increases. However, a saturation point (at least for UCNEs, VISTA enhancers, UTRs and CCDS) seems to emerge at point $W = 3000nt$, and it has been adopted as the default window length for gwRVIS calculation, also to avoid reducing gwRVIS's resolution ability by selecting even longer windows.

Considering a fixed window length $W = 3000nt$, we also performed sensitivity analysis for the MAF threshold, to characterize variants as either common or rare (Supplementary Fig. 1b). We tested eight MAF values regarding the AUC score achieved for intergenic versus non-intergenic regions classification: 0.01%, 0.05%, 0.1%, 0.5%, 1%, 5%, 10%. We observe a similar performance peak at MAF values of 0.1% and 0.5%. Eventually, we adopted as our default value MAF = 0.1% (rather than 0.5%) to be able to detect the effect of even more rare variants.

**Score stability across differing human reference cohort samplings.** To evaluate the robustness of a statistical metric it is important to show that the score distribution is not highly sensitive to the reference dataset that was adopted to construct the metric. Although we defined the gwRVIS adopted throughout this paper based on the TOPMed dataset ($n = 62,784$ individuals; Freeze5 release), we also reconstructed the score using a smaller human reference cohort from the gnomAD WGS dataset ($n = 15,708$ individuals; gnomAD version r2.1.1), following the same preprocessing with TOPMed, where applicable (see Methods). We observe that the global gwRVIS window distribution is highly correlated between the two datasets (Pearson's $r$: 0.91; $p$ value $< 2.2 \times 10^{-16}$; Fig. 3b). As cohort sizes increase we anticipate that the reference datasets will have sufficient resolution to allow for opportunities to shorten the window sizes and also reduce the definition of common variants from the current 1 in 500 humans (autosomal MAF of 0.1%) to less frequent, but still not uncommon carrier rate of 1 in 5000 humans (MAF of 0.01%).

**Exploration of heteroskedasticity during gwRVIS construction**. The gwRVIS score is provided by the studentized residuals for linear regression, modeling common vs. all variants. A scatter plot of the two plotting all gene windows shows that there is some degree of heteroskedasticity, with a variance of common variants increasing as a function of all variants (Supplementary Fig. 15a). We sought to explore if accounting for heteroskedasticity significantly changes the resulting set of gwRVIS scores. Thus, we additionally considered weighted ordinary least squares (WOLS) as a means to correct for lower gwRVIS variance in windows with lower numbers of all variants. WOLS provides a direct means to model heteroskedasticity through a weight matrix, the inverse of which is directly proportional to the covariance of the error terms. Here, we assume the weight matrix is diagonal, with values provided by the inverse of all variants.

By applying the WOLS regression model we observe that heteroskedasticity has been remedied (Supplementary Fig. 15b–d). No significant difference can be observed, however, between the two versions of gwRVIS, as Pearson's correlation between the two is $r = 0.96$ ($p < 1 \times 10^{-308}$). In addition, residuals from windows with more variants seem to be considerably shrunk compared to those from windows with very few variants (Supplementary Fig. 15d). This discrepancy may amplify signals from noisier windows (with fewer variants) and penalize windows of potentially greater confidence (with more variants). Thus, we have selected to model gwRVIS with the non-weighted version of linear regression, also as a direct extension of previously published methods[6].

**gwRVIS vs. Orion benchmarking in protein-coding regions**. We compared gwRVIS to Orion[9], another method that looks into intolerance to variation of noncoding regions by employing the site frequency spectrum. We benchmarked the gwRVIS distribution across the two sets of CCDS and non-CCDS regions (996 and 989 regions, respectively) that were originally adopted by the Orion paper, mapping into 12,260 and 2601 nonoverlapping gwRVIS-derived 3 kb windows, respectively. We benchmarked the gwRVIS predictive power against Orion on the same regions (Orion score median $-0.0448$ vs. $-0.118$, Mann–Whitney $p$ value $= 0.001$). When adopting the gwRIVS for the same comparisons, the separation of the two score distributions was considerably greater (Supplementary Fig. 3; gwRVIS median $-0.436$ vs. $-0.201$; Mann–Whitney $p$ value $= 8.1 \times 10^{-23}$ using all CCDS sites). Because of the variability in the number of windows for the two regions, we also repeated the experiment using a CCDS sample of equal size to the non-CCDS set and the significant departure between the two distributions persisted (gwRVIS median $-0.468$ vs. $-0.201$; Mann–Whitney $p$ value $= 8.44 \times 10^{-15}$).

**Noncoding variants annotation used during training and benchmarking**. We compiled a list of pathogenic variants by retaining all ClinVar variants labeled as "Pathogenic" or "Likely_pathogenic" in the clinical significance ("CLNSIG") field. The set of benign variants is the set of all unique variants in denovo-db that are annotated as "control" in their primary phenotype.

**gwRVIS and external genome-wide scores training**. All genome-wide scores used during benchmarking (except for JARVIS) were trained using a simple Logistic Regression model with fivefold cross-validation, given an inverse regularization strength $C = 1$, "lbfgs" as the optimization algorithm and a maximum number of iterations for optimization convergence max_iter=10,000. The external benchmarked genome-wide scores are CADD, CDTS, DANN, LINSIGHT, ncER, phastCons46way, phyloP46way, and Orion. All scores used during benchmarking have single-nucleotide resolution and so the score from the respective nucleotide referenced by each variant is retained for the classification task. All three scores per nucleotide represented by CADD and DANN have been taken into consideration to capture the average variation profile for each position. Orion scores were extracted based on the gnomAD WGS dataset (15,708 individuals, version 2) using a window length of 1000nt. For each score, we compiled a set of pathogenic variants defined in each genomic class, wherever there is a value defined for that score. A benign set of equal size has been randomly subset from the entire set of benign variants compiled from denovo-db and used alongside the respective pathogenic set for the cross-validation training (score distributions across pathogenic and the full benign sets are shown for gwRVIS and another four genome-wide scores in Supplementary Fig. 5). All logistic regression models with cross-validation were fit using scikit-learn (v. 0.23.2).

**JARVIS training with deep learning**. JARVIS was trained using four different models, one with Gradient Boosting and three Deep Learning-based:

a) a Gradient Boosting classifier with n_estimators=100 trees, max_features=5 and max_depth=2 (to reduce risk of over-fitting).

b) a feed-forward DNN with two hidden layers, having 128 nodes each (a rectified linear unit or "relu" has been used as the activation function for all hidden nodes while "softmax" with two nodes has been employed in the output layer to provide the probability scores for each of the predicted classes).

c) a CNN with two sets of Convolution-MaxPooling layers (both 1-dimensional), followed by two fully connected layers, with 64 and 128 nodes (the parameters for the two convolutional layers are: filters = 64, strides = 2, kernel_size = 11, padding = "valid", and filters = 64 strides = 2, kernel_size = 3,

padding = "valid", respectively, while for both MaxPooling layers are: pool_size = 4 and strides = 2) and a "softmax" output layer with two nodes.

d) a combined DNN and CNN, with the same architecture as described in (b) and (c) (without the "softmax" output layers), concatenated and further processed by another two fully connected layers with 64 and 128 nodes, respectively, again ending with a "softmax" output layer with two nodes.

The Jarvis deep learning parameter selection was based on previously published methods that have fine-tuned neural net-associated parameters for inputs of similar type and size[33]. Specifically, JARVIS adopts the following parameters: depth of network (2 layers), filter size ($k = 11$ and 3 in 1st and 2nd CNN layer, respectively), learning rate of 0.0001 with an Adam optimizer, and early stopping after 10 epochs of the validation loss not getting improved. Furthermore, dropout layers were introduced after every max-pooling and/or fully connected layer for regularization purposes (dropout-ratio: 0.2), to avoid overfitting on the training set.

The JARVIS Gradient Boosting model was fit using scikit-learn (v. 0.23.2) while all neural network-based JARVIS models were trained using tensorflow (v. 1.14.0).

**Raw sequences used as features in deep learning**. Windows sequences with ambiguous nucleotides (i.e., "N"s) have been removed from training and the final predictions (a total of 286 windows). A window length of 3 kb, the same as the one used for gwRVIS construction, has been employed to extract sequences surrounding each nucleotide variant position. One-hot encoding has been applied to each sequence, representing each nucleotide with a binary vector of length 4 with a single non-zero value, specifically: A = [1, 0, 0, 0], T = [0, 0, 0, 1], G = [0, 0, 1, 0] and C = [0, 1, 0, 0].

**Sequence-derived and functional annotation features**. The sequence-derived features adopted are the 7-mer (heptamer) mutability rate[43], CpG di-nucleotides (custom defined as consecutive CG di-nucleotides, e.g., CG and CGCG), and GC content within the 3 kb tiled mutually exclusive windows. The external genomic annotation used is based on Ensembl (GRCh37; release 75) on CTCF binding sites, enhancers, open chromatin, transcription factor binding sites, and histone marks that overlap with each genomic window used for the construction of gwRVIS, based on a representative blood cell-line annotation (CD14+ monocytes). These annotations were extracted via the Ensembl BioMart portal, (Ensembl Regulation 101 resource), and have also been made available in the JARVIS GitHub repository.

**Statistical significance assessment of predictive performance**. DeLong's test was performed for every cross-validation classification task to assess the significance of differences in the predictive performance of the benchmarked genome-wide scores. Two one-sided tests were performed for each pair of compared scores, estimating the significance of JARVIS being significantly better or worse, respectively, in each case. All results from DeLong tests are available in Supplementary Tables 1–18.

**Motif analysis from CNNs**. We trained JARVIS using exclusively the CNN module. Upon training, we scan each sequence from the pathogenic training set using the filters learned by JARVIS (filter length $k = 11$) and capture the activation sum across each sub-sequence. We then focus on all 11-mers that have achieved an activation sum 0.9 times higher than the max. activation sum. We perform multiple sequence alignment across the subset of the selected most-activated sequences using cd-hit[44] (v. 4.8.1), to identify the predominant clusters of 11-mers that were learned by the CNN's 1st layer (cd-hit alignment threshold: 0.8; word size: 7). The identified clusters were then fed to TomTom[45] (v. 5.2.0) against the eukaryote, JASPAR-vertebrate, and Uniprobe-mouse databases of known motifs available within the MEME Suite of tools[46] (Supplementary Data 1). We could then identify the most significantly aligned JARVIS sequence clusters and most frequently hit known motifs (Supplementary Figs. 22 and 23), of which the intersection was visualized employing sequence logos, generated by TomTom (Supplementary Fig. 24).

**Accounting for redundancy of CNN-learnt sequence patterns**. As we infer the feature contribution of CNN models trained with input sequences, it is likely that identical or almost identical sequences get assigned to different Position Weight Matrices.

We aimed to account for the redundancy of identical sequences by performing multiple-sequence alignment of JARVIS-learnt patterns prior to aligning them with known motifs. This is performed using the cd-hit tool (alignment threshold: 0.8; word size: 7). This step doesn't completely preclude the possibility of some clusters having identical sequences, however, we have confirmed that the redundancy is drastically reduced. Specifically, the top 100 reported JARVIS-learnt sequence clusters (Supplementary Data 1) contain only 2 consensus sequences that are present in another cluster as well.

In general, we can observe that the same JARVIS cluster (identified uniquely by its "JARVIS_Cluster_ID"; Supplementary Data 1) can map to several TomTom known motifs ("TomTom_Target_ID" column; Supplementary Data 1). That explains why the same JARVIS consensus sequence repeated multiple times in the motif analysis: in most cases, the same JARVIS-learnt cluster has mapped significantly to multiple motifs from TomTom (Supplementary Fig. 22). In other cases (Supplementary Fig. 23), the reported consensus sequences refer to known

motifs available in the MEME Suite of tools. So, the presence of identical sequences here reflects the redundancy of known motifs present in the "EUKARYOTE/ jolma2013", "JASPAR/JASPAR2018_CORE_vertebrates_non-redundant" and "MOUSE/uniprobe_mouse" databases which have been used for the known motif alignments. Finally, in Supplementary Fig 24, we observe again a redundancy in the known motifs space, while the JARVIS-learnt patterns stacked underneath each of them do not have identical consensus sequences and usually differ to a few nucleotides positions in most cases.

**Benchmarking of structural variants**. Each structural variant spans across multiple nucleotides, from two base pairs up to a few million (Supplementary Fig. 12). In order to capture the aggregate profile of a score across a structural variant, we calculate a summary statistic comprising four statistics: median, mean, first and third quartiles of the respective genome-wide score. Eventually, we assign to each structural variant the average value of the aforementioned statistics and use these values for the benchmarking classification tasks.

**Reporting summary**. Further information on research design is available in the Nature Research Reporting Summary linked to this article.

## Data availability

Genome-wide JARVIS and gwRVIS scores are publicly available at http://jarvis.public. cgr.astrazeneca.com. Relevant data used for generating JARVIS and gwRVIS are available at the JARVIS GitHub repository (https://github.com/astrazeneca-cgr-publications/ jarvis) and in the following public resources: TOPMed (https://bravo.sph.umich.edu/ freeze8/hg38), GnomAD (https://gnomad.broadinstitute.org/downloads), ClinVar (ftp:// ftp.ncbi.nlm.nih.gov/pub/clinvar), UCNEbase (https://ccg.epfl.ch/UCNEbase), FANTOM5 (https://fantom.gsc.riken.jp/5), denovo-db (https://denovo-db.gs. washington.edu/denovo-db/Download.jsp), and Ensembl annotation via BioMart (https://www.ensembl.org/biomart/martview).

## Code availability

All code for generating JARVIS and gwRVIS as well as for benchmarking purposes are available at a public GitHub repository:

https://github.com/astrazeneca-cgr-publications/jarvis

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

## Acknowledgements

We thank Quanli Wang for their comments on the paper.

## Author contributions

Conception and design of the study: S.P. and D.V. Developed the method and generated the genome-wide scores: D.V. Contributed to methods and analytical strategies: D.V, R.D., and S.P. Performed validation analyses: D.V, R.D., L.M., and A.G. Wrote the paper: D.V. and S.P. All authors provided input and revisions for the final paper.

## Competing interests

D.V., R.D, L.M., and S.P. are employees of AstraZeneca. D.V. and S.P. are shareholders of AstraZeneca. L.M.'s work was funded by the AstraZeneca post-doctorate program. A.G. declares no competing interests.
