## [Peer Review File · Nature Communications]

Reviewer #1 (Remarks to the Author):

In this manuscript, the authors developed a suite of computational methods to prioritize non-coding pathogenic variants. The key innovation is a new constrained score, gwRVIS, defined as the degree of depletion of common variants in each sliding window across the genome. The proposed methods have the potential to provide useful information on variant effects that is complementary to existing methods. I have some comments detailed below.

Major comments:

1. Differences in AUROCs have been used to support the advantages of the authors' methods. However, the authors did not provide p-values for these differences. Thus, it is unclear to me whether the authors' methods are statistically significantly better than alternative approaches. Following a common practice in statistics, the authors should use DeLong's test to compute a p-value whenever two AUROCs are compared, and discuss the statistical significance of their findings accordingly.
2. Figure 2a in the main text shows that gwRVIS has limited power to identify constrained regions when the number of variants is small. It may be explained by the heteroscedasticity in the data, as evident by the increase of residual variance with the total number of variants in Figure 2a. I suggest the authors fit a linear regression model with heteroscedasticity to the same data, and define a new RVIS score as the ratio of the residual and the estimated standard deviation from the heteroscedastic model. This modification will increase the variance of the RVIS score when the estimated standard deviation is small, and hence mitigate the undesired dependence of gwRVIS score on the total number of variants.
3. The authors used non-coding pathogenic variants from ClinVar and putatively benign variants from denovo-db as positive and negative data, respectively. Because most ClinVar variants are proximal to protein-coding genes, a machine learning method could "cheat" in the prediction of pathogenic variants by using features correlated with distances to protein-coding genes. While the authors didn't use distances to protein-coding genes as a feature in their models, I expect that gwRVIS scores will, nevertheless, be correlated with the distances due to its large window-size (3kb). To fully rule out the impact of the distances to variant prediction, it is better to compare different methods on a set of positive and negative data with matched distances to protein-coding genes. This strategy was originally used in the GWAVA paper (PMID: 24487584).
4. In the prioritization of pathogenic structural variants (SVs), the authors considered SVs in intergenic regions as benign and SVs in other regions as pathogenic. Thus, the author's model should be interpreted as a tool to discriminate between genic and intergenic SVs, not a tool for unbiased prioritization of pathogenic SVs. The authors should revise this part of their manuscript accordingly.
5. I am wondering what convolutional neural networks (CNNs) learned from raw sequences in this paper. If the CNNs learned meaningful patterns for variant prioritization, their convolutional filters should correspond to motifs for transcription factor binding. The authors can use existing methods (e.g., PMID: 26213851 & 27896980) to visualize their convolutional filters and check if they correspond to any known motifs.
6. The authors should also compare their methods with two other tools, EIGEN (PMID: 26727659) and DeepSEA (PMID: 26301843), which I found highly competitive in predicting non-coding pathogenic variants.
7. It is unclear to me if the authors used any regularization techniques, such as early stopping, weight decay, etc., in the training of deep neural networks. If not, these models may suffer from severe overfitting.

Reviewer #2 (Remarks to the Author):

In their manuscript "Prioritization of non-coding regions based on human genomic constraint and primary sequence context with deep learning", Vitsios et al. present a measure of a genome-wide windowed variant intolerance score (gwRVIS) and an integration of this measure with other annotations in a pathogenicity score (JARVIS) applicable for non-coding sequences. The work highlights that the derived pathogenicity score does not use species conservation and may therefore have advantages over previous measures. While the authors do not explicitly prove such an advantage, they highlight the performance of their score in a number of validation and test data sets. The work is interesting and relevant for a larger audience; however, I have a number of comments and major concerns about the applied cross-validation approach.

Major comments:

- ClinVar pathogenic variants correspond to rather small numbers and are highly clustered. While the small numbers are an issue for training complex models (like DNNs), the clustering is an issue for assessing model performance. The authors cannot use a simple cross-validation scheme under these circumstances. They need to make sure that all variants from the same genomic region are in the same fold. Otherwise, their results will be overly optimistic. While this is critical, it does not seem that the authors have done that.
- You are independently optimizing MAF and length parameters. I would expect an interaction as larger windows contain more variants and increase the likelihood of observing more rare alleles. This would require something like a grid-search. Given the rather flat distributions, this effect might be small and I understand if the authors do not want to change the current implementation. However, the authors should mention this limitation and provide the MAF value that they used for optimizing the window size (lines 527-538).
- DNN architecture: You are using very little data to train the DNNs while having several fully connected layers. Have you considered things like drop-out? How did you identify the architectures and how did you optimize hyperparameters like learning rates and number of iterations? If derived from other literature, provide references and values.
- When using the CNN, do you see better base pair level resolution of your scores? What is the resolution? Are your scores useful for fine mapping applications or even pinpointing TF binding sites (probably not)? Do you have means of checking that (e.g. publicly available saturation mutagenesis MPRA data)?

Minor comments:

- Line 75: You are submitting to a journal with a broad readership; consider introducing "studentized residual" at first mention as residual divided by an estimate of its variance. Please note that you are using both "studentized" and "studentised" in your manuscript.
- Line 88: Discuss known effects of GC content
- Lines 91-92/474ff and Fig 1: Provide numbers/proportion of genome left.
- Line 124: UCNE abbreviation used before it is introduced below
- Lines 190-191: FANTOM5 is more than just lncRNAs, please rephrase.
- Line 298: Other than ascertainment bias (with respect to conservation in ClinVar), there is no "overfitting" for LINSIGHT, CADD or DANN. I think the problem is the use of the word "overfitting" by the authors (see below)
- Suppl. Fig 8: Is this the mean ROC or is it the ROC of the mean of the model predictions?
- Lines 298/305/309/364...: The term overfitting is not used correctly. Overfitting describes the lack of generalization of a learner when applied outside of its training data set. This is seen by a reduced performance on a test set. What you mean is giving high weights to certain features, potentially due to an ascertainment bias in the training data sets; thereby showing an increased performance. You need to rephrase. In some sentences you want to say that performance estimates are overly optimistic because test and training overlap, in others you want to talk about biases/being geared towards/having dispositions/preferences... .

- Fig 4: Feature importance highlights features related to genes (CpG islands/GC/H3K36me) what are you making out of that?
- Lines 410/603ff: The authors approach for scoring SVs is similar to SVscore (Ganel L. et al. Bioinformatics 2017) / an extension there-of; it is also related to the LINSIGHT/CADD approach applied in Abel HJ et al Nature 2020. Why are you not including CADD here, you included it in the other comparisons.
- Lines: 482: Provide versions for TOPMed and gnomAD releases
- Lines 555-556: Provide version numbers for all tools (where applicable)
- Line 588: Provide numbers as: X out of Y
- Line 595: Where does the 7-mer mutability rate come from?
- Line 598: How where these annotations obtained? BIOMART/API? Did you download annotation files?

Reviewer #1:

1. Differences in AUROCs have been used to support the advantages of the authors' methods. However, the authors did not provide p-values for these differences. Thus, it is unclear to me whether the authors' methods are statistically significantly better than alternative approaches. Following a common practice in statistics, the authors should use DeLong's test to compute a p-value whenever two AUROCs are compared, and discuss the statistical significance of their findings accordingly.

We have now performed DeLong's tests for every classification task where AUCs are reported. Additionally, we have added four supplementary figures (Suppl. Fig. 16, 19, 20 & 21) as well as a supplementary file (Suppl. File 1) with the results of all performed DeLong's tests. We discuss all associated results in the corresponding main text (lines 228-234; 237-239; 308-312; 325-328; 333-337; 402-415; 464-471) and added a section in Methods describing the analysis procedure (lines 665-670).

In the original training set, JARVIS performs significantly better ($p < 2.7 \times 10^{-4}$) than all other scores that are not impacted by ClinVar-informed data leakage or integrating TSS distance information (ncER and LINSIGHT) in the training set except for DeepSEA which performs comparably with AUC=0.945 over 0.940 for JARVIS (Fig. 4; DeLong $p=0.064$). In the training set that we've added with matched pathogenic and benign variants, based on their distances to closest TSS, JARVIS performs significantly better than all other scores (AUC=0.800; Suppl. Fig. 18; DeLong test $p < 0.0037$, Suppl. Fig. 19b & Suppl. File 1) except for ncER, which has a significantly better performance (AUC=0.87; DeLong test $p=4.9 \times 10^{-5}$), but over-optimistic due to having being pre-trained on a subset of ClinVar pathogenic variants.

With regards to the four validation sets on single-nucleotide variants, JARVIS performed significantly better than all other scores in the GWAS dataset and in the other three JARVIS performed either significantly better from the rest or significantly comparable than a smaller subset of the benchmarked scores (Suppl. Fig. 20). Finally, on the structural variant validation sets, JARVIS significantly outperforms all other scores in 6 out of 8 datasets (DeLong test $p < 0.022$, Suppl. Fig. 21 & Suppl. File 1), except for LINSIGHT in the inversion-span Structural Variant test set, with results from both being comparable (AUC=0.751 vs 0.747, from JARVIS and LINSIGHT, respectively; DeLong test $p < 0.087$). In the remaining two SV classes JARVIS outperforms some scores (ncER, eigenPC) or is significantly outperformed by others in one case (LINSIGHT, Orion and CADD - Suppl. Fig. 13a, 21 & Suppl. File 1).

2. Figure 2a in the main text shows that gwRVIS has limited power to identify constrained regions when the number of variants is small. It may be explained by the heteroscedasticity in the data, as evident by the increase of residual variance with the total number of variants in Figure 2a. I suggest the authors fit a linear regression model with heteroscedasticity to the same data, and define a new RVIS score as the ratio of the residual and the estimated standard deviation from the heteroscedastic model. This modification will increase the variance of the RVIS score when the estimated standard deviation is small, and hence mitigate the undesired dependence of gwRVIS score on the total number of variants.

We have now considered an alternative version of gwRVIS that considers the heteroscedasticity observed in the original formulation for genomic windows with very low number of variants in general. We have added a supplementary figure that illustrates the heteroscedasticity effect on the original gwRVIS calculation and how this is accommodated for by the weighted regression modeling (Suppl. Fig. 15).

We sought to explore if accounting for heteroskedasticity significantly changes the resulting set of gwRVIS scores. Thus, we additionally considered weighted ordinary least squares (WOLS) as a means to correct for lower gwRVIS variance in windows with lower numbers of all variants. Overall no significant difference can be observed between the two versions of gwRVIS, as Pearson's correlation between the two is $r=0.96$ ($p<1\times 10^{-308}$). Additionally, residuals from windows with more variants seem to be considerably shrunk compared to those from windows with very few variants (Suppl. Fig. 15d), potentially amplifying signals from noisier windows (with fewer number of variants) over windows of potentially higher confidence (with more variants). Thus, we decided to retain the gwRVIS modeling with the non-weighted version of linear regression and added a section in Supplemental Methods describing this approach (Suppl. Methods - lines 3-20).

3. The authors used non-coding pathogenic variants from ClinVar and putatively benign variants from denovo-db as positive and negative data, respectively. Because most ClinVar variants are proximal to protein-coding genes, a machine learning method could “cheat” in the prediction of pathogenic variants by using features correlated with distances to protein-coding genes. While the authors didn't use distances to protein-coding genes as a feature in their models, I expect that gwRVIS scores will, nevertheless, be correlated with the distances due to its large window-size (3kb). To fully rule out the impact of the distances to variant prediction, it is better to compare different methods on a set of positive and negative data with matched distances to protein-coding genes. This strategy was originally used in the GWAVA paper (PMID: 24487584).

We have prepared an alternative version of the training set (referred to as the matched training set) by selecting control variants from denovo-db with similar distribution of TSS distances to closest genes, as compared to the pathogenic variants employed in the JARVIS training set (Suppl. Fig. 17). JARVIS performs significantly better than all other scores (AUC=0.800; Suppl. Fig. 18; DeLong test $p<0.0037$, Suppl. Fig. 19b & Suppl. File 1) except for ncER, which has a significantly better performance (AUC=870; DeLong test $p=4.93\times 10^{-5}$) but boosted by the issue of data leakage. We have added a section in main text describing this additional analysis (lines 319-328).

4. In the prioritization of pathogenic structural variants (SVs), the authors considered SVs in intergenic regions as benign and SVs in other regions as pathogenic. Thus, the author's model should be interpreted as a tool to discriminate between genic and intergenic SVs, not a tool for unbiased prioritization of pathogenic SVs. The authors should revise this part of their manuscript accordingly.

We re-phrased the respective section in main text (lines 445-450) as well as the legend in Fig. 6 to reflect that this benchmarking refers to classification of genic vs non-genic SV effects, with the former class expected to be enriched for clinically relevant SVs.

5. I am wondering what convolutional neural networks (CNNs) learned from raw sequences in this paper. If the CNNs learned meaningful patterns for variant prioritization, their convolutional filters should correspond to motifs for transcription factor binding. The authors can use existing methods (e.g., PMID: 26213851 & 27896980) to visualize their convolutional filters and check if they correspond to any known motifs.

We have performed motif analysis on the sequences learnt by the JARVIC CNN module, based on a similar approach followed in another multi-module deep learning network (Angermueller et al., 2017; PMID: 28395661). Specifically, in order to capture the learnt sequence patterns, we look into the most highly activated filters that comprise the feature maps at the output of the first convolutional layer. We selected

the most activated k -mers (where $k=11$, equal to the selected filter size) in the training set of pathogenic sequences (those that had activation sum 0.9 times higher than the max. activation sum). We then performed multiple sequence alignment between the most-activated sequences with cd-hit, and identified the most predominant clusters of 11-mers that were learnt by the CNN's 1st layer. The identified clusters were then fed to the TomTom web-server against the eukaryote, JASPAR-vertebrate and uniprobe-mouse databases of known motifs (Suppl. File 2). We observed that, despite the fairly small size of the training set, JARVIS CNN module has managed to learn dozens of sequence patterns that align with known vertebrate, human or mouse motifs, many of which are enriched for long Cytosine stretches (Suppl. Fig. 22, 23, 24 & Suppl. File 2). We describe this analysis and results in a section in the main text (lines 359-367) and in Methods (lines 672-683), as well as provide the sequence logs and/or consensus sequences of the top learnt motifs in Suppl. Fig. 22, 23, 24 & Suppl. File 2.

6. The authors should also compare their methods with two other tools, EIGEN (PMID: 26727659) and DeepSEA (PMID: 26301843), which I found highly competitive in predicting non-coding pathogenic variants.

We have now added Eigen-PC to our benchmarking (Ionita-Laza et al. 2016). In certain benchmarks, Eigen-PC did not have sufficient number of annotated nucleotide positions and thus has been omitted from those specific plots (e.g. the matched by TSS distance training set, the GWAS and mendelian validation sets). We found that Eigen-PC performed consistently lower than JARVIS in all applicable tests. DeepSEA was also added across all benchmarks for single-nucleotide variants and performed either significantly lower than JARVIS or in some cases comparably with no significant improvement based on DeLong's test. We could not install the stand-alone version of DeepSEA and the web-server accepts inputs of up to 50,000 data points, thus we were not able to extract DeepSEA scores for the structural variants and so this score is not included in the SV benchmarking.

7. It is unclear to me if the authors used any regularization techniques, such as early stopping, weight decay, etc., in the training of deep neural networks. If not, these models may suffer from severe overfitting.

We have used dropout as our regularization technique across all modules of the deep neural network. Specifically, a dropout layer was introduced after every max-pooling and/or fully-connected layer to avoid overfitting on the training set (dropout-ratio: 0.2). These parameters can be found at the 'feedf_dnn()' and 'cnn2_fc2()' functions at the respective module in GitHub repository (https://github.com/astrazeneca-cqr-publications/JARVIS/blob/master/modules/jarvis/deep_learn_raw_seq/func_api_nn_models.py). We have also added a section in Methods that describes this regularization approach (lines 641-643).

Reviewer #2:

Major comments:

- ClinVar pathogenic variants correspond to rather small numbers and are highly clustered. While the small numbers are an issue for training complex models (like DNNs), the clustering is an issue for assessing model performance. The authors cannot use a simple cross-validation scheme under these circumstances. They need to make sure that all variants from the same genomic region are in the same fold. Otherwise, their results will be overly optimistic. While this is critical, it does not seem that the authors have done that.

We have now added an alternative version of the cross-validation training set, which uses stratification by chromosome during the fold splits. That ensures that variants from the same genomic region cannot be part of both the training and test sets at any cross-validation step, thus removing any bias from data circularity that is highlighted by the Reviewer. JARVIS performance dropped marginally (AUC=0.929, Suppl. Fig. 18) but remained significantly higher than the others (DeLong test $p < 5.5 \times 10^{-11}$; Suppl. Fig. 19c & Suppl. File 1) except for DeepSEA and CADD that performed comparably (AUC=0.922 and 0.913, DeLong test $p = 0.065$ and 0.16 , respectively) and ncER and LINSIGHT that performed significantly better (AUC=0.980 and 0.969, DeLong test $p = 1.2 \times 10^{-8}$ and 2.0×10^{-4} , respectively). We have added a section in the main text describing this analysis and respective results (lines 329-337).

- You are independently optimizing MAF and length parameters. I would expect an interaction as larger windows contain more variants and increase the likelihood of observing more rare alleles. This would require something like a grid-search. Given the rather flat distributions, this effect might be small and I understand if the authors do not want to change the current implementation. However, the authors should mention this limitation and provide the MAF value that they used for optimizing the window size (lines 527-538).

We thank the reviewer for this remark. In order to reduce the parameter search space, we have indeed performed a two-step optimization approach instead of a full-scale grid search. We initially decided on a MAF value of 0.1% as a nominal threshold for common variant annotation. This threshold is consistent with our original implementation of RVIS (Petrovski et al., 2013) and has also been selected in cohort studies like in Cirulli et al., 2020, as a useful threshold for defining rare variants (equivalent to being observed once among 1,000 individuals). By using this nominal MAF value we first optimised the window length parameter and then moved to optimise separately the MAF value, which again converged on 0.1%. We have added a section in Methods describing this analysis and respective results (lines 576-580).

- DNN architecture: You are using very little data to train the DNNs while having several fully connected layers. Have you considered things like drop-out? How did you identify the architectures and how did you optimize hyperparameters like learning rates and number of iterations? If derived from other literature, provide references and values.

As per previous response to Reviewer 1, we have used dropout as our regularisation technique across all modules of the deep neural network. Specifically, a dropout layer was introduced after every max-pooling and/or fully-connected layer to avoid overfitting on the training set (dropout-ratio: 0.2). These parameters can be found at the 'feedf_dnn()' and 'cnn2_fc2()' functions at the respective module in GitHub repository (https://github.com/astrazeneca-cgr-publications/JARVIS/blob/master/modules/jarvis/deep_learn_raw_seq/func_api_nn_models.py).

The JARVIS deep learning parameter selection was based on a previously published successful method that fine-tuned parameters for inputs of similar size (Angermueller et al., 2017; PMID: 28395661). Specifically, the derived parameters that are employed by JARVIS, include the depth of the network (2

layers), filter size ($k=11$ and 3 in each layer), learning rate ($lr=0.0001$) with an Adam optimizer and early stopping after 10 epochs of the validation loss not getting improved. We have added a section in methods explaining the selection of parameters used by the JARVIS neural network (lines 637-643).

- When using the CNN, do you see better base pair level resolution of your scores? What is the resolution? Are your scores useful for fine mapping applications or even pinpointing TF binding sites (probably not)? Do you have means of checking that (e.g. publicly available saturation mutagenesis MPRA data)?

The JARVIS model that was trained exclusively with CNNs on raw genomic sequences performed lower than the model employing a feed-forward neural network applied to structured data (like gwRVIS, functional genomic annotations, etc. (Suppl. Fig. 8)). The combination of the two has also proven to boost predictive performance in the majority of validation sets, which reflects that the CNN module has actually learnt additional patterns from the raw genomic sequences.

In order to examine if JARVIS' CNN module learnt any biologically meaningful motifs, we performed motif analysis based on a similar approach followed in another multi-module deep learning network (Angermueller et al., 2017; PMID: 28395661). Specifically, in order to capture the learnt sequence patterns, we look into the most highly activated filters that comprise the feature maps at the output of the first convolutional layer. We selected the top 90th percentile of most activated k -mers (where $k=11$, equal to the selected filter size) in the training set of pathogenic sequences. We then performed multiple sequence alignment between the most-activated sequences with cd-hit, and identified the most predominant clusters of 11-mers that were learnt by the CNN's 1st layer. The identified clusters were then fed to the TomTom web-server against the eukaryote, JASPAR-vertebrate and uniprobe-mouse databases of known motifs (Suppl. File 2). We observed that, despite the fairly small size of the training set, JARVIS CNN module has managed to learn dozens of sequence patterns that align with known vertebrate, human or mouse motifs, many of which are enriched for long Cytosine stretches (Suppl. Fig. 22, 23, 24 & Suppl. File 2). We describe this analysis and results in a section in the main text (lines 359-367) and in Methods (lines 672-683), as well as provide the sequence logs and/or consensus sequences of the top learnt motifs in Suppl. Fig. 22, 23, 24 & Suppl. File 2.

Minor comments:

- Line 75: You are submitting to a journal with a broad readership; consider introducing "studentized residual" at first mention as residual divided by an estimate of its variance. Please not that you are using both "studentized" and "studentised" in your manuscript.

We have now added an expanded description of the studentised residual term the first time we introduce it. We have also adopted the "studentized" spelling throughout.

- Line 88: Discuss known effects of GC content

We have added a brief description of known GC effects in main text (lines 89-91).

- Lines 91-92/474ff and Fig 1: Provide numbers/proportion of genome left.

We have added a reference to the total genomic region of higher confidence (~75% of the genome) that resulted after all filtering steps (lines 532-534).

- Line 124: UCNE abbreviation used before it is introduced below

We have added the full name of UCNE at the first mention.

- **Lines 190-191: FANTOM5 is more than just lncRNAs, please rephrase.**

We rephrased the FANTOM5 definition: "FANTOM5 is a resource that contains mammalian promoters, enhancers, lncRNAs and miRNAs, including a collection of nearly 20,000 functional lncRNAs in human." (lines 190-192).

- **Line 298: Other than ascertainment bias (with respect to conservation in ClinVar), there is no "overfitting" for LINSIGHT, CADD or DANN. I think the problem is the use of the word "overfitting" by the authors (see below)**

We have re-phrased accordingly in main text to reflect the data leakage and/or ascertainment bias problem in each case (lines 302-304; 313-315; 328).

- **Suppl. Fig 8: Is this the mean ROC or is it the ROC of the mean of the model predictions?**

Mean ROC in all these plots refers to the ROC of the mean of model predictions from each cross-validation split. We have added this clarification at Suppl. Fig. 8's legend.

- **Lines 298/305/309/364...: The term overfitting is not used correctly. Overfitting describes the lack of generalization of a learner when applied outside of its training data set. This is seen by a reduced performance on a test set. What you mean is giving high weights to certain features, potentially due to an ascertainment bias in the training data sets; thereby showing an increased performance. You need to rephrase. In some sentences you want to say that performances estimates are overly optimistic because test and training overlap, in others you want to talk about biases/being geared towards/having dispoitions/preferences...**

We have re-phrased accordingly in main text to reflect the data leakage and/or ascertainment bias problem in each case (lines 302-304; 313-315; 328).

- **Fig 4: Feature importance highlights features related to genes (CpG islands/GC/H3K36me) what are you making out of that?**

It's been recently reported that CpG islands in promoter regions have been associated with loss-of-function intolerance of proximal coding regions (Boukas et al., 2020), thus JARVIS may be able to preferentially prioritise non-coding variants that have a direct functional effect on coding regions. Additionally, the presence of CpG dinucleotides has been reported as the primary indicator of local mutability (Havrilla et al. 2018). We discuss this accordingly in main text (lines 353-356).

- **Lines 410/603ff: The authors approach for scoring SVs is similar to SVscore (Ganel L. et al. Bioinformatics 2017) / an extension there-of; it is also related to the LINSIGHT/CADD approach applied in Abel HJ et al Nature 2020. Why are you not including CADD here, you included it in the other comparisons.**

We have added CADD as part of the SV benchmarks (Fig. 6, Suppl. Fig. 13, 21).

- **Lines: 482: Provide versions for TOPMed and gnomAD releases**

We have now added versions of each resource (line 528).

- Lines 555-556: Provide version numbers for all tools (where applicable)

We have added the versions for the libraries used to fit the models (**lines 644-645**).

- Line 588: Provide numbers as: X out of Y

We added a reference to a supplementary figure (Suppl. Fig. 12) that shows the distribution of lengths across the different SV classes (**line 687**).

- Line 595: Where does the 7-mer mutability rate come from?

We thank the reviewer for pointing this unintentional omission out - we have now added the relevant citation (Aggarwala & Voight, 2016 - **line 656**)

- Line 598: How where these annotations obtained? BIOMART/API? Did you download annotation files?

We extracted these annotations via the Ensembl BioMart portal (Ensembl Regulation 101 resource) and have made the relevant version of the downloadable files available at the JARVIS GitHub repo as well (https://github.com/astrazeneca-cqr-publications/JARVIS/tree/master/ensembl/GRCh37-Regulatory_Features). We have added a description in the Methods (**lines 661-663**).

Reviewer #1 (Remarks to the Author):

The authors have addressed most of my concerns. I only have a few relatively minor comments on the revised manuscript.

Line 75: Studentized residual is defined as the residual divided by an estimate of standard deviation, not variance. I think the authors mistakenly used "variance" instead of "standard deviation" in this sentence.

Line 307: I agree that supervised machine learning models trained on known pathogenic variants could suffer from data leakage if the distance to the nearest TSS is used as an input feature. On the other hand, CADD, DANN, and LINSIGHT were trained on polymorphisms in healthy individuals rather than known pathogenic variants, so these methods should not suffer from data leakage. I suggest the authors tone down this part of the manuscript accordingly.

Line 333: I don't understand why ncER is subject to data leakage when pathogenic and benign variants are matched by the distance to the nearest TSS. The authors argued that data leakage is due to the use of the distance to the nearest TSS but it has been controlled for in this analysis, so it shouldn't be a concern here. Again, I suggest the authors tone down this part of the manuscript accordingly.

Reviewer #2 (Remarks to the Author):

Vitsios et al. have provided a revision of their manuscript "Prioritization of non-coding regions based on human genomic constraint and primary sequence context with deep learning", where they present a measure of a genome-wide windowed variant intolerance score (gwRVIS) and an integration of this measure with other annotations in a pathogenicity score (JARVIS) applicable for non-coding sequences. They have addressed comments from two reviewers with several additional analyses. I have only some minor comments left, which I trust with the authors and editors to decide whether those actually need to be addressed.

Suppl. Fig 6: You are ordering the panels by "class separation performance" without actually giving this measure in the plot. I would recommend to provide this metric, e.g. in the title of each panel.

Fig 3/Suppl. Fig 18: Most prominent in the left panel of Fig S18 you can see pronounced steps in the AUC curves. Those are typically caused by missing or identical values. Do you know what is causing them here?

Suppl. Fig 22-24: You have some identical sequences that you are assigning different PWMs. It would be nice to work on clusters of PWMs or to somehow reduce the redundancy otherwise. Overall, I have to admit that I would have used an existing framework for this analysis like DeepExplainer (Lundberg and Lee 2017; Lundberg et al. 2020; Avsec et al. 2019b) or the TF Modisco package (<https://github.com/kundajelab/tfmodisco>) – even though the analysis of the filters is legitimate and we have done that in our own work too. To me it seems clear that one would always match some motifs, however it is unclear to me whether this is more than some spurious hits for a network that has learned to identify GC rich sequences. Other than comparing to equally abundant and GC-matched kmers the human reference, I do not have a good idea how to assess the significance of these hits.

Reviewer #1:

1. Line 75: Studentized residual is defined as the residual divided by an estimate of standard deviation, not variance. I think the authors mistakenly used “variance” instead of “standard deviation” in this sentence.

We thank the reviewer for pointing out this oversight. This has been corrected in the main text (line 67).

2. Line 307: I agree that supervised machine learning models trained on known pathogenic variants could suffer from data leakage if the distance to the nearest TSS is used as an input feature. On the other hand, CADD, DANN, and LINSIGHT were trained on polymorphisms in healthy individuals rather than known pathogenic variants, so these methods should not suffer from data leakage. I suggest the authors tone down this part of the manuscript accordingly.

The comment around the potential data leakage issue referred exclusively to ncER (which has also been trained with a set of ClinVar pathogenic variants), however, we agree it was not clear in the text. We have now edited this statement to clearly refer only to ncER (lines 254-255 and 270-272). We have edited accordingly the comments about the other scores to highlight only the potential ascertainment bias issue due to the variant distribution of the JARVIS training set with respect to closest TSS (lines 259-262 and 270-272).

3. Line 333: I don't understand why ncER is subject to data leakage when pathogenic and benign variants are matched by the distance to the nearest TSS. The authors argued that data leakage is due to the use of the distance to the nearest TSS but it has been controlled for in this analysis, so it shouldn't be a concern here. Again, I suggest the authors tone down this part of the manuscript accordingly.

The data leakage issue refers exclusively to the fact that ncER's training set includes a set of known pathogenic variants from ClinVar (and HGMD), as mentioned in the text (lines 253-254). Thus, ncER is tested in that case on a variant set (the JARVIS training set) which, at least partially, overlaps with its own training set, indicating a data leakage between the training and test set. The matching of pathogenic and benign variants with respect to TSS distance resolves another issue, which is the bias that most of the known ClinVar pathogenic variants are close to protein-coding genes, as mentioned in the text (lines 257-259).

Reviewer #2:

1. Suppl. Fig 6: You are ordering the panels by "class separation performance" without actually giving this measure in the plot. I would recommend to provide this metric, e.g. in the title of each panel.

We mention in the Suppl. Fig. 6's legend that plots are sorted based on AUC performance. The plots themselves depict the distribution of each score (density) in two sets of variants (pathogenic or benign) and the AUC metric has been used solely for their relative placement in the figure's layout. We have now added a title in Suppl. Fig. 6 for the whole figure emphasizing this:

“Genome-wide scores distribution between pathogenic and benign variant sets (ordered from left to right in decreasing order of AUC performance with 5-fold cross-validation)”

2. Fig 3/Suppl. Fig 18: Most prominent in the left panel of Fig S18 you can see pronounced steps in the AUC curves. Those are typically caused by missing or identical values. Do you know what is causing them here?

We have examined a subset of scores to explore what is causing these ‘steps’ in AUCs and have observed both the existence of missing and identical values, in different scores/settings. For example, Orion’s “long horizontal step” in Fig. 3 is due to a high degree of missing values, while nCER’s flat vertical increases are mostly attributed to data points receiving the same nCER score. So, we can’t conclude comprehensively for a single cause but can comment on the existence of both missing and identical values causing these formations in the AUC plots.

3. Suppl. Fig 22-24: You have some identical sequences that you are assigning different PWMs. It would be nice to work on clusters of PWMs or to somehow reduce the redundancy otherwise. Overall, I have to admit that I would have used an existing framework for this analysis like DeepExplainer (Lundberg and Lee 2017; Lundberg et al. 2020; Avsec et al. 2019b) or the TF Modisco package (<https://github.com/kundajelab/tfmodisco>) – even though the analysis of the filters is legitimate and we have done that in our own work too. To me it seems clear that one would always match some motifs, however it is unclear to me whether this is more than some spurious hits for a network that has learned to identify GC rich sequences. Other than comparing to equally abundant and GC-matched kmers the human reference, I do not have a good idea how to assess the significance of these hits.

We have tried to account for the redundancy of identical sequences by performing multiple-sequence alignment of JARVIS-learned patterns prior to aligning them with known motifs. As mentioned in methods section (lines 640-643), this is performed with cd-hit (alignment threshold: 0.8; word size: 7). Although this step doesn’t completely preclude the possibility of some clusters having identical sequences, we have confirmed that the redundancy is drastically reduced. In particular, Suppl. File 2 contains the top 100 most prominent JARVIS-learned sequence motifs, with only 2 of them being present in another cluster.

In the same file, we can observe that the same JARVIS cluster (identified uniquely by its ‘JARVIS_Cluster_ID’) is mapping to several TomTom known motifs (‘TomTom_Target_ID’ column). This is primarily why in Suppl. Fig. 22 we observe the same JARVIS consensus sequence repeated multiple times: in most cases, the same JARVIS-learned cluster has mapped significantly to multiple motifs from TomTom.

With regards to Suppl. Fig. 23, the reported consensus sequences refer to known motifs available in the MEME Suite of tools. So, the presence of identical sequences here reflects the redundancy of known motifs present in the “EUKARYOTE/jolma2013”, “JASPAR/JASPAR2018_CORE Vertebrates_non-redundant” and “MOUSE/uniprobe_mouse” databases which have been used for these alignments.

Finally, in Suppl. Fig 24, we observe again a redundancy in the known motifs space, while the JARVIS-learned patterns stacked underneath each of them do not have identical consensus sequences and usually differ to a few nucleotide positions in most cases.

In general, we agree with the reviewer that identifying the significance of the learned motifs and their true biological impact is still an active area of research. We have now added a section in Methods to clarify the cause of redundancy encountered in each case (Methods: lines 649-669).